# VISUAL PROMPT TUNING FOR TEST-TIME DOMAIN ADAPTATION

## ABSTRACT

Models should be able to adapt to unseen data during test-time to avoid performance drops caused by inevitable distribution shifts in real-world deployment scenarios. In this work, we tackle the practical yet challenging test-time adaptation (TTA) problem, where a model adapts to the target domain without accessing the source data. We propose a simple recipe called *Data-efficient Prompt Tuning* (DePT) with two key ingredients. First, DePT plugs visual prompts into the vision Transformer and only tunes these source-initialized prompts during adaptation. We find such parameter-efficient finetuning can efficiently adapt the model representation to the target domain without overfitting to the noise in the learning objective. Second, DePT bootstraps the source representation to the target domain by memory bank-based online pseudo-labeling. A hierarchical self-supervised regularization specially designed for prompts is jointly optimized to alleviate error accumulation during self-training. With much fewer tunable parameters, DePT demonstrates not only state-of-the-art performance on major adaptation benchmarks VisDA-C, ImageNet-C, and DomainNet-126, but also superior data efficiency, i.e., adaptation with only 1% or 10% data without much performance degradation compared to 100% data. In addition, DePT is also versatile to be extended to online or multi-source TTA settings.

## 1 INTRODUCTION

Deep neural networks achieve excellent performance when the testing data (target domain) follow the same distribution as the training data (source domain). However, their generalization ability on the testing data is not guaranteed when a distribution shift occurs between the source and the target. Even simple domain shifts, like common corruptions (Hendrycks & Dietterich, 2019) or appearance variations (Geirhos et al., 2018), can lead to a significant performance drop. Solving the problem of domain shift is non-trivial. On one hand, it is impossible to train a single model to cover an infinite number of domains. On the other hand, training individual models for each domain requires lots of annotated samples, which induces significant data collection and labeling costs.

In this paper, we tackle the practical yet challenging test-time domain adaptation (TTA) problem. Compared with conventional unsupervised domain adaptation (UDA) (Long et al., 2015), TTA adapts the source domain initialized model with the unlabeled target domain data during testing without access to source data. TTA avoids the privacy issue of sharing the source data and is desirable for real-world applications. We focus on both offline and online TTA settings. For offline adaptation, also known as source-free adaptation (Kundu et al., 2020; Liang et al., 2020), the model is first updated with unlabeled target data and then inference. For online adaptation, the model keeps updating and inferencing at the same time, given the testing data that comes in a stream.

The key challenges of TTA lie in two folds. First, how to effectively modulate the source domain initialized model given a noisy unsupervised learning objective? Tent (Wang et al., 2020) optimizes the parameters of batch normalization layers, which is parameter-efficient but lacks adaptation capacity. On the other hand, SHOT (Liang et al., 2020) tunes the feature encoder; AdaContrast (Chen et al., 2022) trains the whole model. Given the current over-parameterized models, these methods are prone to overfitting to the unreliable unsupervised learning objective, especially when the amount of target domain data is limited. We present more evidences in Appendix A to illustrate our motivation to use visual prompt tuning. Second, given only unlabeled target domain data, use

what kind of learning objective for optimization? Existing works propose to use unsupervised objectives, including entropy minimization (Wang et al., 2020), self-supervised auxiliary task (Sun et al., 2019b), pseudo labeling (Lee et al., 2013), or a combination of the above objectives (Liang et al., 2020; Chen et al., 2022; Liu et al., 2021b). However, these unsupervised objectives are either not well aligned with the main task (Liu et al., 2021b), or give noisy supervision (Liang et al., 2020).

In this work, we propose **D**ata-**e**fficient test-time adaptation via **P**rompt **T**uning (DePT). Our recipe is simple yet effective, where two key ingredients are proposed to address the two challenges mentioned above, see in Fig. 2. First, a collection of learnable visual prompts (Jia et al., 2022) are trained with labeled source data alongside the vision Transformer (Dosovitskiy et al., 2020). In the adaptation phase, we only finetune the prompts and the classification head, while freezing the backbone parameters. Although the knowledge learned in the source domain is retained in the unchanged backbone, tuning solely the prompts can effectively modulate the model for target domain adaptation. Second, for the learning objective given only unlabeled target domain data, DePT bootstraps the source domain initialized model via self-training, where the pseudo labels are first predicted and then refined by soft voting from nearest neighbor data points in a memory bank. To further alleviate the accumulation of errors during self-training, we design a hierarchical fine-grained self-supervised regularization term for the prompts to encourage better target domain representation learning. These two objectives are complementary and jointly optimized.

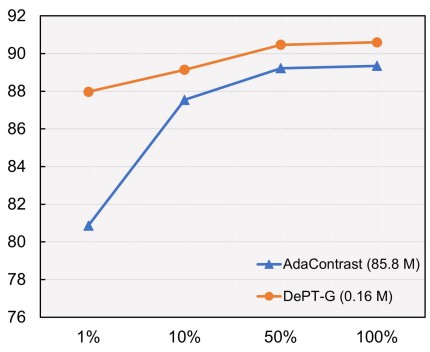

Figure 1: The test-time adaptation performance of different methods with respect to the data ratio on the VisDA dataset. The number in the legend denotes the number of tunable parameters. DePT-G outperforms previous SOTA AdaContrast on 100% target data with only 0.19% tunable parameters. The superiority of DePT is more significant on the low data settings.

The two ingredients in DePT offer multiple merits and lead to state-of-the-art performance on major domain adaptation benchmarks: VisDA-C (Peng et al., 2017), ImageNet-C (Hendrycks & Dietterich, 2019) and DomainNet-126 (Peng et al., 2019). Tuning prompts during test-time is much more parameter-efficient than full fine-tuning. As illustrated in Fig. 1, with only 0.19% tunable parameters, DePT achieves 1.2% higher performance than previous state-of-the-art AdaContrast (Chen et al., 2022). Moreover, parameter efficiency brings data efficiency to DePT. Compared with AdaContrast, DePT has significant improvement in the low data regime. For example, with only 1% of unlabeled target domain data in VisDA-C, DePT achieves 88.0% average accuracy, which surpasses AdaContrast by 7.1%. Under the more challenging online TTA setting, DePT demonstrates superior performance with 85.9% average accuracy on VisDA-C. For robustness to corruptions, DePT achieves the least error rate on 14 out of 15 types of corruptions on the severity level of 5 on ImageNet-C with a 34.9% average error rate. Besides, we also show that DePT is flexible to extend to more DA scenarios like multi-source (Peng et al., 2019) domain adaptation.

## 2 RELATED WORK

**Unsupervised domain adaptation (UDA)** setting is commonly used, where unlabeled data from the testing distribution (target domain) is available during training along with the labeled data from the training distribution (source domain). Many studies attempt to solve this problem using style transfer (Hoffman et al., 2018; Tang et al., 2021; Taigman et al., 2016), feature alignment (Long et al., 2015; Sun et al., 2017; Peng et al., 2019) or learning domain-invariant feature via adversarial training (Ganin et al., 2016; Tzeng et al., 2017). However, the UDA setting has a strong assumption that the source and target domain data are both accessible during training, which is not always true as it is difficult to access source data after the model is deployed. Moreover, these methods usually need to retrain the whole framework if new domains come.

**Test-time domain adaptation (TTA)** is a more challenging and practical setting where no source domain data is available during adaptation. Such property makes TTA avoid the privacy issue and transmission burden of source data sharing, and enables the pipeline of training the model once and adapting the model to any unknown test distributions. The offline TTA is essentially source-free domain adaptation (Kundu et al., 2020; Liang et al., 2020), where the model can update several epochs on the unlabeled target domain data before inferencing. The online TTA setting requires the model to keep updating and inferencing at the same time batch-by-batch as long as there are testing data. Some recent works explore the TTA setting with different unsupervised objectives and model modulation methods. Test-time training (TTT) (Sun et al., 2019b) introduces a self-supervised rotation prediction task as the test-time optimization objective. Tent (Wang et al., 2020) modulates the batch normalization parameters by minimizing entropy. SHOT (Liang et al., 2020) adapts the feature extractor to the target domain with a combination of information maximization and pseudo labeling. SHOT++ (Liang et al., 2021) further applies a semi-supervised learning step to propagate the label of high-confidence samples to low-confidence samples. HDMI (Lao et al., 2021) also optimizes for information maximization, but introduces a disparity regularization on multiple hypotheses. Ada-Contrast (Chen et al., 2022) tunes the whole model with memory bank-based pseudo labeling and contrastive loss. DePT has several critical differences from these approaches. Most of these methods are proposed for ConvNet, while we switch to vision transformer. For model modulation, we freeze the backbone parameters and learn the target-specific prompts. For the learning objective, we jointly optimize an online memory bank pseudo labeling loss and a hierarchical self-supervised loss specially designed for the prompts. DePT is also flexible on a variety of DA scenarios.

**Self-supervised learning** methods have made great progress in unsupervised visual representation learning. Early attempts use pretext tasks, e.g. colorization (Zhang et al., 2016), rotation prediction (Gidaris et al., 2018), jigsaw (Noroozi & Favaro, 2016), as the self-supervised objective. Another large body of works focus on discriminative approaches (Chen et al., 2020; He et al., 2020), named contrastive learning, which push the representation of positive pairs closer and spread negative pairs further. Recent works show that with a carefully designed structure or learning objective, unsupervised features can be learned without discrimination between instances (Grill et al., 2020; Caron et al., 2021). Besides unsupervised visual representation learning, researchers also explore applying self-supervised objectives for model robustness (Hendrycks et al., 2019) and domain adaptation (Sun et al., 2019a;b). Inspired by BYOL (Grill et al., 2020) and DINO (Caron et al., 2021), we propose a hierarchical self-supervised regularization that are jointly optimized with pseudo labels.

**Prompt tuning** originates from NLP. Prompts are text instructions that are prepended to the input as a hint to inform the pre-trained language model about the task (Brown et al., 2020). Besides text prompts that are mostly constructed via heuristics, soft prompts, i.e., continuous vectors that are optimized via gradient descent, achieve comparable performance with full fine-tuning but have much less tunable parameters (Li & Liang, 2021; Liu et al., 2021a). Prompt tuning is also explored in vision tasks. BRS (Jang & Kim, 2019) perturbs the input for prediction refinement. L2P (Wang et al., 2022) is proposed for continual learning. Bahng et al. (2022) proposed to pad prompts in the pixel space to adapt large-scale models in vision. VPT (Jia et al., 2022) is a highly related work proposed for transfer learning. VPT inserts randomly initialized prompts to the pre-trained model, then optimizes them with the downstream task's label. DAPL (Ge et al., 2022) introduced prompt learning to the UDA setting, yet it is highly dependent on the large-scale vision-language model (Radford et al., 2021). In contrast, DePT doesn't require a text encoder, and the visual prompts learned with the source domain data are updated with unlabeled target domain data.

## 3 METHOD

We address the close-set test-time domain adaptation (TTA) setting where source data is not accessible during adaptation. In source domain training, $n_s$ labeled samples $\{x_s^i, y_s^i\}_{i=1}^{n_s}$ from the source domain $\mathcal{D}_s$ are given, where $x_s^i \in \mathcal{X}_s$ and $y_s^i \in \mathcal{Y}_s$ are data and the corresponding labels. A model has a general architecture with parameters $\theta$ trained to learn the function $f_s : \mathcal{X}_s \to \mathcal{Y}_s$. We assume the model is parameterized by a neural network and consists of two modules: the feature encoder $g_s \colon \mathcal{X}_s \to \mathbb{R}^d$ and a classifier $h_s \colon \mathbb{R}^d \to \mathbb{R}^C$, where $d$ is the dimension of the feature and $C$ is the number of classes. During target domain adaptation, unlabeled data $\{x_t^i\}_{i=1}^{n_t}$ from target domain $\mathcal{D}_t$ is available, where $x_t^i \in \mathcal{X}_t$. The goal of TTA is to adapt the source domain trained model $f_s$ to

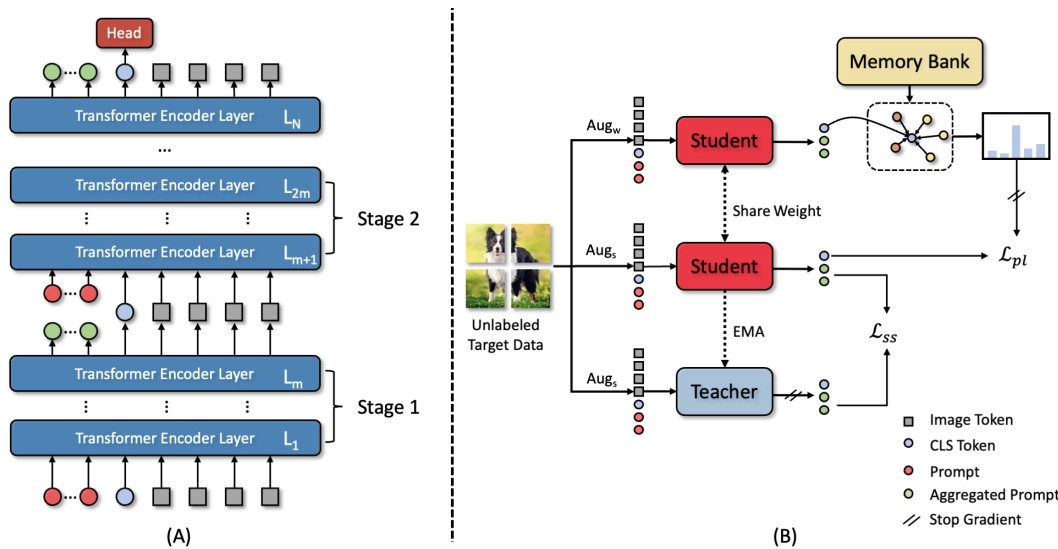

Figure 2: The overview of DePT. (A) We split ViT into multiple stages and prepend prompts to the input of each stage. The prompts, along with the backbone, are initialized with labeled source domain data. Only prompts and the classification head (in red) are finetuned during adaptation, while the backbone is frozen (in blue). (B) The proposed adaptation framework. The pseudo labels of target data are first predicted and then refined by memory bank-based soft voting for self-training. A hierarchical self-supervised objective is proposed to improve target representation and alleviate error accumulation of self-training.

target domain $f_t : \mathcal{X}_t \rightarrow \mathcal{Y}_t$. We assume the target domain share the same label space with source, i.e., $\mathcal{Y}_s = \mathcal{Y}_t = \mathcal{Y}$.

## 3.1 VISUAL PROMPT TUNING

How to modulate the model during TTA is a non-trivial problem. The optimization objective of TTA is unsupervised and usually noisy, making full parameter tuning sensitive, easy to diverge, and prone to overfitting to the noise, especially when the number of target data is limited. On the other hand, tuning little parameters has limited adaptation capacity. Therefore, we propose to adopt visual prompt tuning to solve this dilemma. We provide more evidence and motivation for prompt tuning in Appendix A.

Visual prompts are a set of learnable continuous parameters that are prepended to the input of Transformer layers, see in Fig. 2 (A). For a Vision Transformer (ViT), the input image is first divided into patches and then embedded into $d$-dimensional image tokens $\boldsymbol{E}_0$. Similar to Jia et al. (2022), we concatenate the prompts with CLS token and image tokens to the input of Transformer layers after the image embedding layer. In order to adjust the adaptation capacity, DePT inserts new prompts at different levels of Transformer layers. To be specific, we split the Transformer layers into $M$ stages, where each stage has $m = N/M$ layers. A collection of $p$ prompts $\boldsymbol{P}$ are inserted into the first layer of each stage by concatenating with the image and CLS tokens. We select $M = \{1, 4, 12\}$ to form three variants of DePT: DePT-Shallow (S), DePT-Group (G) and DePT-Deep (D). The prompt augmented ViT is formulated as:

$$
\begin{aligned}
[\boldsymbol{c}_i, \boldsymbol{Z}_i, \boldsymbol{E}_i] &= L_i([\boldsymbol{c}_{i-1}, \boldsymbol{P}_{i-1}, \boldsymbol{E}_{i-1}]) && i = 1, m+1, 2m+1, \ldots, N-m \\
[\boldsymbol{c}_i, \boldsymbol{Z}_i, \boldsymbol{E}_i] &= L_i([\boldsymbol{c}_{i-1}, \boldsymbol{Z}_{i-1}, \boldsymbol{E}_{i-1}]) && i = else \\
\boldsymbol{y} &= head(\boldsymbol{c}_N)
\end{aligned}
\tag{1}
$$

where $\boldsymbol{c}_i$ and $\boldsymbol{Z}_i$ represent the CLS token and the aggregated prompt features computed by the $i$-th Transformer layer, respectively. By altering the number of stages $M$ and the number of prompts $p$, we can easily control the number of tunable parameters for different adaptation capacity.

The prompt augmented ViT model learns the function $f_s : \mathcal{X}_s \rightarrow \mathcal{Y}_s$ by labeled source data with cross-entropy loss, where all parameters, including the prompts and the backbone, are optimized. In the test-time adaptation phase, we initialize the target model with the source model's weights. We only fine-tune the parameters of the prompts and the classification heads, while all other parameters are fixed. The knowledge learned in the source domain is retained in the frozen backbone. The prompts are finetuned to learn the target-specific knowledge to adjust the representations non-linearly through self-attention. The classification head is trained to learn a new decision boundary.

## 3.2 LEARNING OBJECTIVE

Our learning objective consists of two parts. First, an online pseudo labeling with a memory bank refinement mechanism is used to bootstrap the source domain learned representation to the target domain. Second, we design a self-supervised objective on the CLS token and the aggregated prompt to form a hierarchical regularization to further reduce the noise in the pseudo labels and improve the target representation quality.

### 3.2.1 PSEUDO LABELING

Pseudo labeling has widely been used in semi/self-supervised learning. Inspired by Tarvainen & Valpola (2017); Sohn et al. (2020); Chen et al. (2022), we use a student-teacher model and an online memory bank refinement to generate pseudo labels. To be specific, see in Fig. 2 (B), the student $f_t$ and the teacher $f'_t$ share the same architecture, and are both initialized with the source model's weight $\theta_s$ at the beginning of adaptation. The teacher is not updated by gradients, but by exponential moving average (EMA) from the student's weights. The teacher maintains an memory bank that stores the feature embeddings and predicted probabilities of target samples. The memory bank is online updated as a first-in-first-out queue from training mini-batches. Given an unlabeled target domain data $x_t$, a weak and a strong random image transformation: $\mathcal{T}_w$ and $\mathcal{T}_s$ are applied. The embedding of the weakly augmented view is first obtained from the student. The pseudo label is then obtained by a soft voting, i.e. average of the probability, from the top-$k$ nearest neighbors in the memory bank: $\hat{y}_t = \arg\max \mathcal{V}(f_t(\mathcal{T}_w(x_t)))$, where $\mathcal{V}$ denotes the soft voting with memory bank. The student is trained to enforcing the prediction for the strong-augmented view to match $\hat{y}_t$ with a cross-entropy loss:

$$\mathcal{L}_{pl} = \frac{1}{n_t} \sum_{i=1}^{n_t} H(\hat{y}_t, f_t(\mathcal{T}_s(x_t))), \tag{2}$$

where $H(a, b) = -a \log b$. The details can be found in Appendix B.1. Intuitively, the student is fast updating during training, making the pseudo label produced by the student to be noisy. The EMA updated teacher can be seen as an ensemble of the student along training. Thus the memory bank maintained by the teacher stores more stable and high-quality predictions. The soft voting further leverages nearby data point cluster to produce more accurate pseudo labels.

### 3.2.2 HIERARCHICAL SELF-SUPERVISED REGULARIZATION

Due to the domain shift, the model trained on the source domain cannot directly produce good representations of the target data, i.e., the representation of images from the same class may spread apart. Although using the student-teacher model and the memory bank refinement can significantly improve the quality of pseudo-labels, they still cannot completely avoid the accumulation of errors during self-training. Therefore, we design a hierarchical self-supervised regularization for the proposed prompt augmented ViT to improve the model's representation of target data.

For the prompt augmented ViT, the CLS token of the output of ViT encoder $c_N$ serves as the holistic image representation for classification. The prompts work similarly to the CLS token: prepending to the input of Transformer layers. With self-attention, the prompts can inject domain-specific information into other tokens, and aggregate instance features from other tokens. Each prompt will attend to different attributes and features of the image. The prompts at the output of the Transformer layer, which we call aggregated prompts, serve as a fine-grained representation of the image. Moreover, the aggregated prompts from different stages may attend to distinctive levels of features and thus form a hierarchical representation. Inspired by Caron et al. (2021) and Grill et al. (2020), given different views of the same image, we push both their holistic feature representations (CLS token) and their hierarchical fine-grained representations (aggregated prompts) closer.

We use the DINO (Caron et al., 2021) framework here with a few key modifications. The student encoder $g_t$ and EMA teacher encoder $g'_t$ are reused with the ones in the pseudo labeling section. A series of projection heads $q_t \colon \mathbb{R}^d \to \mathbb{R}^K$ are introduced to project the CLS token $c_t$ or the aggregated prompts $Z_i$ of the target sample $x_t$ into $K$ dimensions, where each stage of the prompt augmented ViT has its own projection head. Two random strong-augmented views of the target sample is generated. One is passed to the student, and the other one is passed to the teacher. The projected CLS token and aggregated prompts are obtained from the student: $\tilde{c}_t = q_t(c_t)$, $\{\tilde{Z}_i = q_t(Z_i)\}_{i \in \mathcal{I}}$ and similarly for the teacher $\tilde{c}'_t$, $\{\tilde{Z}'_i\}_{i \in \mathcal{I}}$, where $\mathcal{I} = \{m, 2m, \dots, N\}$ is the index of the last Transformer layer in each stage. For the holistic regularization on the CLS token, we apply the DINO loss. The probability distribution of the student output is obtained by applying a softmax function: $P_t(x_t) = \mathrm{softmax}(\tilde{c}_t)$, and so is the teacher's $P'_t(x_t)$. The student aims at matching the teacher's predicted distribution by minimizing the cross-entropy loss:

$$\mathcal{L}_{ss\_cls} = H(P'_t(x_t), P_t(x_t)). \tag{3}$$

The technique of centering and sharpening of DINO are also used to avoid collapse, where the details can be found in Appendix B.2. For the hierarchical fine-grained regularization, we minimize the mean squared error between the aggregated prompts of the student and teacher:

$$\mathcal{L}_{ss\_prompt} = \frac{1}{M} \sum_{i \in \mathcal{I}} \sum_{j=1}^{p} \left( 2 - 2 \cdot \frac{\langle \tilde{Z}_i^j, \tilde{Z}_i^{j'} \rangle}{||\tilde{Z}_i^j||_2 \cdot ||\tilde{Z}_i^{j'}||_2} \right). \tag{4}$$

In addition, although the regularization is applied on the aggregated prompts one by one, i.e. the $i$-th aggregated prompt of the student is pushed closer to the $i$-th one of the teacher, we find the model tend to learn the trivial solution where all aggregated prompt become similar. To avoid this, we add a diversity term to encourage different prompts to attend to different features, thereby enabling them to learn rich target-specific knowledge. Specifically, we maximize the cosine distance among the $p$ aggregated prompts of the student:

$$\mathcal{L}_{ss\_div} = \frac{1}{M} \sum_{i \in \mathcal{I}} \sum_{j=1}^{p} \sum_{k=j}^{p} \left( 1 - \frac{\langle \tilde{Z}_i^j, \tilde{Z}_i^k \rangle}{||\tilde{Z}_i^j||_2 \cdot ||\tilde{Z}_i^k||_2} \right), \tag{5}$$

where $j$ and $k$ are the index of the aggregated prompts at the output of the $i$-th Transformer layer. Therefore, DePT minimizes the following loss function for TTA:

$$\mathcal{L} = \alpha \mathcal{L}_{pl} + \beta_1 \mathcal{L}_{ss\_cls} + \beta_2 \mathcal{L}_{ss\_prompt} - \lambda \mathcal{L}_{ss\_div}. \tag{6}$$

## 4 EXPERIMENTS

### 4.1 EXPERIMENTAL SETUP

We evaluate the effectiveness and versatility of DePT in a variety of TTA scenarios, covering three major domain adaptation benchmarks. **VisDA-C** (Peng et al., 2017) is a large-scale domain adaptation benchmark that focuses on 12-class synthesis-to-real object recognition task. We use the training set as the source and the validation set as the target. The source domain contains 152k synthetis images generated by rendering 3D models while the target domain has 55k real images sampled from Microsoft COCO. **ImageNet-C** (Hendrycks & Dietterich, 2019) is a large-scale benchmark to evaluate models' robustness against 15 types of corruptions. **DomainNet-126** (Peng et al., 2019) provides images from multiple domains for classification task. Since the original DomainNet dataset has noisy labels, we follow the previous work (Saito et al., 2019; Chen et al., 2022) to use a subset of DomainNet, which contains 126 classes from 4 domains (Real, Sketch, Clipart, Painting).

**Models** Different from previous works that use ConvNet as the backbone, e.g. ResNet26/50/101, we use ViT-B as a strong backbone network for adaptation. We use the model and ImageNet pre-trained weight from the timm library (Wightman, 2019) in our experiments.

**Baselines** We compare our method with both classical UDA methods and test-time adaptation methods. For UDA methods, we compare to DANN (Ganin & Lempitsky, 2015), CDAN (Long et al., 2018), CAN (Kang et al., 2019), SWD (Lee et al., 2019) and MCC (Jin et al., 2020). Note that all UDA methods need access to both source and target domain data, while ours doesn't. For TTA methods, we compare with Tent (Wang et al., 2020), SHOT (Liang et al., 2020), CFA (Kojima et al., 2022) and AdaContrast (Chen et al., 2022).

Table 1: Classification accuracy (%) on VisDA-C train → val adaptation. Upper table: UDA methods. Middle table: TTA methods. Bottom table: online setting. All models with ViT-B backbone are reproduced by us, while the results with R101 backbone are cited from the original paper. Bold and underline denote the first and second high results.

| Method | Backbone | plane | bcycl | bus | car | horse | knife | mcycl | person | plant | sktbrd | train | truck | Avg. |
|---|---|---|---|---|---|---|---|---|---|---|---|---|---|---|
| DANN | R101 | 81.9 | 77.7 | 82.8 | 44.3 | 81.2 | 29.5 | 65.1 | 28.6 | 51.9 | 54.6 | 82.8 | 7.8 | 57.4 |
| CDAN | R101 | 85.2 | 66.9 | 83.0 | 50.8 | 84.2 | 74.9 | 88.1 | 74.5 | 83.4 | 76.0 | 81.9 | 38.0 | 73.9 |
| CAN | R101 | 97.0 | 87.2 | 82.5 | 74.3 | 97.8 | 96.2 | 90.8 | 80.7 | 96.6 | 96.3 | 87.5 | 59.9 | 87.2 |
| SWD | R101 | 90.8 | 82.5 | 81.7 | 70.5 | 91.7 | 69.5 | 86.3 | 77.5 | 87.4 | 63.6 | 85.6 | 29.2 | 76.4 |
| MCC | R101 | 88.7 | 80.3 | 80.5 | 71.5 | 90.1 | 93.2 | 85.0 | 71.6 | 89.4 | 73.8 | 85.0 | 36.9 | 78.8 |
| Source | ViT-B | 99.0 | 69.1 | 78.2 | 71.9 | 87.7 | 58.4 | 96.4 | 35.0 | 52.7 | 92.5 | 96.0 | 17.6 | 71.2 |
| Tent | ViT-B | 99.0 | 76.9 | 79.2 | 80.8 | 93.9 | 84.2 | 95.9 | 54.5 | 74.6 | 92.9 | 95.6 | 22.9 | 79.2 |
| SHOT | ViT-B | **99.5** | 91.8 | 88.7 | 65.1 | 98.6 | 98.0 | 96.0 | 66.1 | 95.1 | **98.9** | 96.8 | 52.4 | 87.3 |
| AdaCon | ViT-B | **99.5** | **94.2** | 91.2 | 83.7 | 98.9 | 97.7 | **96.8** | 71.5 | 96.0 | 98.7 | 97.9 | 45.0 | 89.2 |
| DePT-S (Ours) | ViT-B | 98.7 | 90.1 | 87.4 | 70.7 | 98.9 | 96.0 | **96.8** | 75.2 | 91.9 | 97.9 | 96.2 | 52.5 | 87.7 |
| DePT-G (Ours) | ViT-B | 99.2 | 93.0 | 93.4 | 85.6 | 99.4 | **98.6** | 96.7 | 76.2 | 98.2 | 97.9 | 96.6 | 52.0 | 90.6 |
| DePT-D (Ours) | ViT-B | 99.4 | 93.8 | **94.4** | **87.5** | 99.4 | 98.0 | 96.7 | 74.3 | **98.4** | 98.5 | 96.6 | 51.0 | **90.7** |
| Online Setting | | | | | | | | | | | | | | |
| AdaCon | ViT-B | 95.4 | 75.2 | 83.4 | **64.2** | 95.8 | 93.2 | 92.4 | 65.3 | 92.6 | 82.9 | **95.1** | 39.7 | 81.3 |
| DePT-G (Ours) | ViT-B | **98.3** | **87.6** | **88.9** | 62.1 | **98.4** | **95.2** | **95.9** | **67.1** | **95.2** | **97.3** | 94.9 | **50.1** | **85.9** |

## 4.2 VISDA-C DATASET RESULTS

**DePT demonstrates state-of-the art performance even with much less tunable parameters.** Tab. 1 compares DePT with UDA and TTA methods. Without access to source data, DePT with more strong ViT-B backbone achieves better performance than the UDA methods. Among TTA methods, with much less tunable parameters, DePT-G (0.16M) and DePT-D (0.47M) outperforms the strong baseline AdaContrast (85.8M) by +1.4% and +1.5%. DePT-S, with only 0.048M tunable parameters, still has a huge improvement (+16.5%) compared with no adaptation. Although the backbone is frozen, tuning the prompts can effectively adjust the representation for the target domain.

**DePT gains more advantages under limited data settings.** Previous TTA methods usually assume the target domain data is sufficient for adaptation. However, even without label, obtaining a large amount of target domain data is still expensive in some scenarios. Therefore, we investigate the impact of the target domain data quantity on the performance of the TTA methods by evaluating with 1%, 10%, 50%, and 100% of target data, see in Fig. 3 (A). DePT shows strong data efficiency. DePT-G still has 87.97% average accuracy under 1% data, which is only 2.59% lower than with 100% data. In contrast, AdaContrast's average accuracy is only 80.87% at 1% data, which is 8.48% lower than 100% data. For DePT-D, its adaptation capacity is the highest when target data is abundant, e.g., in 100% data. DePT-D has more decline than DePT-G and DePT-S under 1% data, but is still superior to AdaContrast. For DePT-S, the absolute performance is not high, but it shows the best robustness against data quantity changes, which only decreases 1.17% under 1% data compared to 100% data.

**DePT has strong performance on the more challenging online TTA setting.** In the online setting, the model adapts to target data that comes in batch sequentially in a stream. Each data batch can only be used for training once. We do not apply learning rate decay and turn off the memory bank and directly use the model prediction as the pseudo label. The bottom of Tab. 1 shows the performance of DePT online. DePT achieves 85.9% average accuracy, surpassing AdaContrast by +4.6%.

## 4.3 IMAGENET-C DATASET RESULTS

**DePT is effective on various corruption types.** Tab. 2 shows the top-1 error rate on the highest severity (level-5) corruptions on ImageNet-C. In the offline adaptation, we compare with SHOT-IM (Liang et al., 2020), CFA (Kojima et al., 2022), and AdaContrast (Chen et al., 2022). DePT-G achieves the lowest error for 14 out of 15 types of corruptions. In the online adaptation setting, DePT-G outperforms online AdaContrast and CFA by 3.7% and 1.2%, respectively.

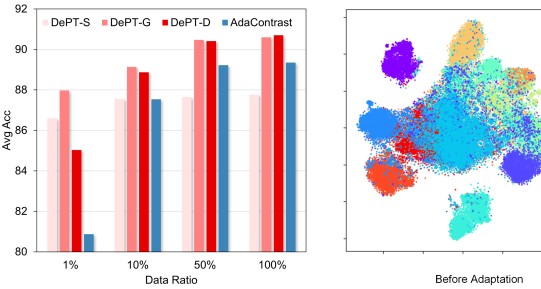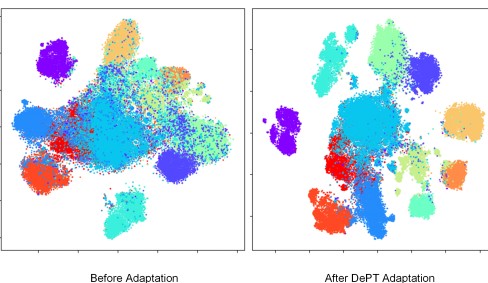

Figure 3: (A) Performance comparison on VisDA-C dataset with different data ratio for TTA. DePT shows less performance degradation when the training data reduces. (B) t-SNE visualization of the DePT-G model before and after adaptation on VisDA-C dataset. Different color denotes classes.

Table 2: Top-1 error rate (%) on 15 corruptions with highest severity (level-5) on ImageNet-C. All models use the ViT-B backbone, where Source, AdaContrast and DePT-G are reproduced by us, while SHOT-IM and CFA are cited from Kojima et al. (2022).

| Method | Gauss | Shot | Impul | Defoc | Glass | Motion | Zoom | Snow | Frost | Fog | Bright | Contr | Elast | Pixel | Jpeg | Avg |
|---|---|---|---|---|---|---|---|---|---|---|---|---|---|---|---|---|
| Source | 52.2 | 51.5 | 51.3 | 55.3 | 69.3 | 50.8 | 55.8 | 46.2 | 50.1 | 45.3 | 25.3 | 68.9 | 54.9 | 34.4 | 33.8 | 49.7 |
| SHOT-IM | **43.4** | 42.3 | 42.4 | 44.1 | 48.5 | 40.9 | 43.9 | 35.3 | 37.4 | 33.8 | 22.4 | 42.0 | 39.2 | 29.8 | 31.5 | 38.4 |
| AdaContrast | 43.6 | 42.2 | 43.5 | 45.7 | 52.0 | 40.2 | 44.1 | 33.2 | 38.1 | 31.7 | 23.3 | 42.9 | 37.1 | 28.5 | 30.8 | 38.4 |
| DePT-G(Ours) | 45.6 | **41.8** | **41.7** | **40.6** | **42.0** | **38.4** | 40.2 | 29.4 | 32.7 | 26.5 | 20.8 | 35.5 | 32.2 | 26.4 | 29.5 | 34.9 |
| *Online Setting* | | | | | | | | | | | | | | | | |
| AdaContrast | 45.6 | 44.2 | 44.2 | 47.5 | 57.8 | 41.3 | 45.7 | 35.4 | 39.9 | 33.6 | 23.2 | 46.3 | 38.3 | **28.1** | 30.4 | 40.1 |
| CFA | **43.1** | **42.0** | **41.9** | 45.6 | 51.1 | 40.1 | 43.4 | 33.6 | 35.9 | 32.3 | **21.0** | 41.2 | 35.7 | 28.3 | **29.8** | 37.6 |
| DePT-G (Ours) | 46.3 | 44.3 | 44.2 | **41.8** | **44.0** | **38.2** | 42.9 | 30.8 | 33.4 | 27.8 | 23.7 | 36.8 | 32.1 | 28.2 | 31.8 | **36.4** |

## 4.4 DOMAINNET DATASET RESULTS

**DePT consistently improves across seven domain shifts.** Tab. 3 shows the comparison on the seven domain shifts in the DomainNet-126 dataset. All three variants of DePT show consistent improvement on all seven domain shifts compared with no adaptation. DePT-G performs the best among them and achieves 81.7% average accuracy. Although the proposed DePT-G and DePT-D have much smaller tunable parameters, they outperform the previous SOTA AdaContrast. In the online setting, DePT achieves 3.1% higher performance than AdaContrast. We also find that models with ViT-B backbone demonstrate stronger robustness against domain shifts than the R50 backbone.

**Beyond one-on-one adaptation.** We further evaluate DePT to multi-source TTA setting, details of the setting can be found in Appendix B.3. See in Tab. 4, DePT makes the best use of multi-source domain knowledge by utilizing a set of domain-specific prompts and a shared domain-agnostic prompt. In contrast, SHOT (Liang et al., 2020) makes one-by-one domain adaptation separately and then makes predictions by an ensemble, which does not jointly optimize for multi-source domains.

## 4.5 ANALYSIS

**Visual prompt tuning helps to learn better target representation**. Fig. 3 (B) shows the t-sne visualization of the target feature embedding before and after adaptation. Before adaptation, the clusters are mixed and difficult to distinguish due to domain shift. After adaptation, the embeddings are more scattered, and the number of samples that fall into the wrong cluster is less. Although the backbone is fixed, the prompts can effectively adapt the representation through attention layers.

**Prompt tuning is highly effective for adaptation**. Tab. 5 shows the impact of the number of prompts. Increasing the number of stages or the number of prompts introduces more tunable parameters. As the VisDA-C validation set has abundant unlabeled data, increasing tunable parameters usually results in better performance, e.g., DePT-G-100 and DePT-D-50 achieve SOTA adaptation performance. Surprisingly, even with one prompt, DePT still achieves decent adaptation accuracy.

Table 3: Classification accuracy (%) of seven domain shifts on DomainNet-126. All models use ViT-B backbone.

| Method | R→C | R→P | P→C | C→S | S→P | R→S | P→R | Avg. |
|---|---|---|---|---|---|---|---|---|
| Source | 68.7 | 75.9 | 69.8 | 67.7 | 74.1 | 60.4 | 86.4 | 71.8 |
| Tent | 69.1 | 77.7 | 70.1 | 67.0 | 75.3 | 61.7 | 88.0 | 72.7 |
| SHOT | 80.2 | 81.5 | 79.8 | 74.2 | **82.2** | 72.8 | 90.3 | 80.1 |
| AdaCon | 81.8 | 81.8 | 82.0 | 75.8 | **82.2** | 73.7 | **90.4** | 81.1 |
| DePT-S | 81.2 | 81.4 | 80.9 | 75.1 | 81.7 | 73.1 | 88.9 | 80.3 |
| DePT-G | **83.3** | 81.6 | 82.6 | **77.8** | 81.3 | **75.1** | 89.9 | **81.7** |
| DePT-D | 82.8 | **82.5** | **82.8** | 76.9 | 81.2 | 74.6 | 89.8 | 81.5 |
| Online Setting | | | | | | | | |
| AdaCon | 72.1 | 77.9 | 71.8 | 70.1 | 75.7 | 65.5 | **87.9** | 74.2 |
| DePT-G | **74.8** | **78.3** | **74.2** | **72.5** | **77.2** | **68.5** | 87.2 | **76.1** |

Table 4: Classification accuracy (%) of multi-source DA setting on DomainNet-126 dataset. $\mathcal{R}$ denotes the **remaining** three domains except the single source. All models use ViT-B backbone

| Multi Source | ERM | SHOT | DePT-G |
|---|---|---|---|
| $\mathcal{R} \to$R | 88.7 | 90.7 | 91.0 |
| $\mathcal{R} \to$C | 76.7 | 83.1 | 83.7 |
| $\mathcal{R} \to$S | 69.2 | 75.9 | 78.3 |
| $\mathcal{R} \to$P | 78.7 | 83.4 | 84.0 |
| Average | 78.3 | 83.2 | 84.2 |

Table 5: Ablation study of DePT variants and prompt number on VisDA-C.

| Prompt Num | 1 | | 10 | | 50 | | 100 | |
|---|---|---|---|---|---|---|---|---|
| Model | Acc | Params/M | Acc | Params/M | Acc | Params/M | Acc | Params/M |
| DePT-S | 83.8 | 0.009 | 86.0 | 0.016 | 87.7 | 0.048 | 87.6 | 0.086 |
| DePT-G | 85.4 | 0.010 | 88.4 | 0.040 | 90.6 | 0.16 | 90.6 | 0.32 |
| DePT-D | 86.3 | 0.018 | 89.0 | 0.10 | 90.7 | 0.47 | 90.4 | 0.94 |

Table 6: Ablation of the contribution of each component to the final performance on VisDA-C. The upper table uses DePT-G with 50 prompts and tunes the prompts. The lower table uses the same model but tunes all parameters including the backbone.

| # | PL | MB | CLS Reg. | Prompt Reg. | Prompt Div. | 100% | 1% |
|---|---|---|---|---|---|---|---|
| 0 | | | | | | 73.2 | 73.2 |
| 1 | ✓ | | | | | 85.6 | 78.4 |
| 2 | ✓ | ✓ | | | | 86.9 | 80.4 |
| 3 | ✓ | ✓ | ✓ | | | 89.3 | 85.6 |
| 4 | ✓ | ✓ | ✓ | ✓ | | 88.2 | 84.6 |
| 5 | ✓ | ✓ | ✓ | ✓ | ✓ | 90.6 | 88.0 |
| Full | ✓ | ✓ | ✓ | ✓ | ✓ | 90.7 | 81.3 |

**Contribution of each loss**. Tab. 6 shows the contribution of each loss under 100% and 1% data on VisDA-C. The pseudo labeling and memory bank refinement can improve the accuracy in the target domain, but the improvement depends on the amount of data. They improve by 12.4% under 100% data, but only improve by 7.2% under 1% data. Adding the self-supervise objective on the CLS token allows the model to learn a better representation of the target domain. We tried to further add hierarchical regularization on the prompts. However, directly adding regularization makes prompts to collapse. By further introducing the diversity term, the final performance is boosted to 90.6% under 100% data, and 88.0% under 1% data. The lower table shows the contribution of prompt tuning. With the same learning objective, tuning all parameters achieves good accuracy under abundant target data (100%), but is prone to overfit under limited data (1%).

## 5 CONCLUSION

We propose a data-efficient test-time adaptation approach via visual prompt tuning. We pretrain a collection of visual prompts alongside the model with labeled source domain data. During adaptation, we only adapt the prompts to learn target-specific knowledge with the backbone fixed. A flexible scheme is proposed to insert a certain number of prompts to the input of each stage of the Transformer. By doing so, we can easily control the adaptation capacity and tunable parameters by altering the number of stages and prompts. An online pseudo labeling with memory bank refinement is jointly learnt with a hierarchical self-supervised objective for test-time prompt tuning. We extensively evaluate the proposed method with VisDA-C, ImageNet-C and DomainNet-126 datasets and show that our simple recipe can lead to SOTA performance in multiple settings and superior data efficiency. In addition, we further verify the effectiveness and flexibility of DePT on online and multi-source adaptation settings.

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

## A    MOTIVATION OF USING PROMPT TUNING

How to modulate the model is vital for the TTA setting. Since only unlabeled target data are available during adaptation, the model can only learn through the unsupervised learning objective. However, current unsupervised learning objectives are not always true and usually contain noises and errors. If there are too many parameters to modulate, the current over-parameterized models are prone to overfit to the noise of the unsupervised objective, especially when the amount of unlabeled target data is limited. On the other hand, if modulating too few parameters, although overfitting will be avoided, the adaptation capacity of the model will be limited.

We perform an experiment to verify our hypothesis. We use AdaContrast (Chen et al., 2022) and Tent (Wang et al., 2020) as examples to modulate the model. Among them, AdaContrast finetunes all parameters of the model. Tent only adjusts the transformation parameters of the BatchNormalization layers, which only contains very few parameters. For ViT, we adjust the transformation parameters of the LayerNorm (LN) layer instead. For these two model modulation approaches, we use the learning objective of AdaContrast for training, i.e., MoCo-based self-supervised combined with pseudo labeling.

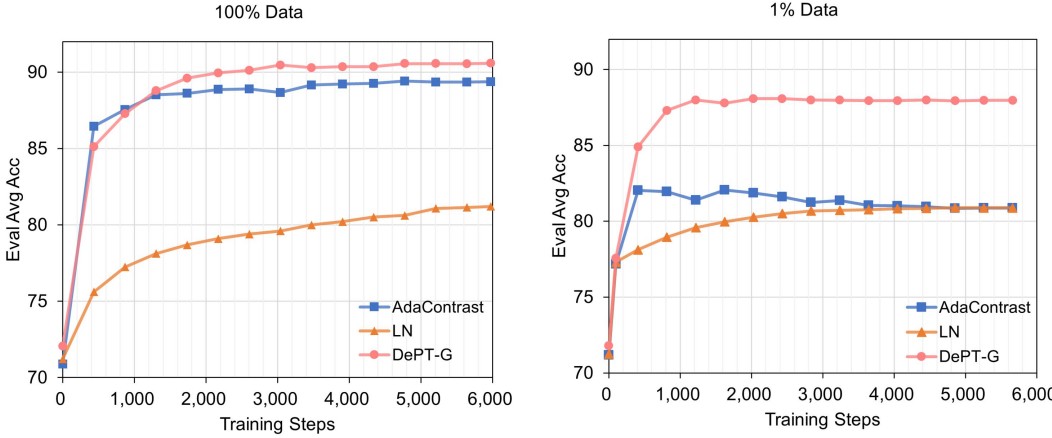

Figure 4: The curve of average accuracy of evaluation v.s. training steps with 100% data and 1% data on VisDA-C dataset.

Fig. 4 shows the results of training the above two modulation approaches with 100% and 1% VisDA-C unlabeled target domain data. Under 100% data, AdaContrast adapts well to the target domain. This is because the target domain of VisDA-C has 55K images, making AdaContrast not easy to overfit. For LN, its performance is limited by the adaptation capacity. Under 1% of the data, with the increasing of training steps, AdaContrast over-fits obviously and the performance of adaptation gradually declines. LN does not overfit but its adaptation performance is still limited by the capacity. In the TTA setting, as no target domain label is available, we currently do not have a good metric to measure when to stop adaptation. Therefore, a method that can avoid over-fitting and have enough adaptation capacity is necessary. Such observation motivate us to explore using visual prompt tuning to solve this dilemma.

## B    IMPLEMENTATION DETAILS

### B.1    PSEUDO LABELING DETAILS

During adaptation, we use a student-teacher model and an online memory bank refinement to generate pseudo labels. The student and the teacher share the same prompt-augmented ViT architecture. They are both initialized with the source model's weight $\theta_s$ at the beginning of adaptation. As the student is fast updating during training, the pseudo label directly produced by the student is usually unstable and noisy. We introduce a slow momentum teacher model whose weights $\theta'_t$ exponential

moving average updated by the student's weights $\theta_t$:

$$\theta'_t = m\theta'_t + (1 - m)\theta_t, \tag{7}$$

where $\theta_t$ is the student's weights, $\theta'_t$ is the teacher's weights, and $m$ is the momentum for updating. The teacher can be seen as an ensemble of student along training, thus is more stable and has higher pseudo label accuracy.

The teacher maintains an online memory bank $B$ with size $N$ to refine the pseudo labels from the student via soft voting, where the memory bank stores the feature embeddings and the predicted probabilities $\{c'_i, p'_i\}_{i=1}^N$ of weakly augmented target data from the teacher. The memory bank $B$ is initialized by randomly selected $N$ samples, and then updated with data from training mini-batch as a first-in-first-out queue.

Given an unlabeled target domain data $x_t$, a weak and a strong random image transformations $\mathcal{T}_w$ and $\mathcal{T}_s$ are applied. The student produces the feature embedding $c_t$ based on the weakly augmented view $\mathcal{T}_w(x_t)$. To refine the prediction of the student, we use $c_t$ to retrieve top-$k$ nearest neighbors in the memory bank $B$ by measuring the cosine distance between $c_t$ and $\{c'_i\}_{i=1}^N$. The refined hard pseudo label is obtained by averaging the probability of the top-$k$ neighbors followed by a $\arg\max$ operation:

$$\hat{y}_t = \arg\max \frac{1}{k} \sum_{i=1}^k p'_i \tag{8}$$

Inspired by FixMatch (Sohn et al., 2020), the student is trained to enforce the cross-entropy loss against the model's output for the strongly-augmented view:

$$\mathcal{L}_{PL} = \frac{1}{n_t} \sum_{i=1}^{n_t} H(\hat{y}_t, f_t(\mathcal{T}_s(x_t))) \tag{9}$$

## B.2 SELF-SUPERVISED OBJECTIVE DETAILS

We use the DINO (Caron et al., 2021) framework for self-supervised learning with a few key modifications. We reuse the student and teacher model in the pseudo labeling with the DINO framework, which are both initialized by the source model. We apply the DINO loss on the CLS token. The sharpening and centering operation is also used to avoid collapse. To be specific, the sharpening operation is applied for both the student and teacher:

$$P_t(x_t)^{(i)} = \frac{\exp(\tilde{c}_t^{(i)}/\tau)}{\sum_{k=1}^K \exp(\tilde{c}_t^{(k)}/\tau)} \tag{10}$$

The teacher is further applied a centering operation:

$$P'_t(x_t)^{(i)} = \frac{\exp((\tilde{c}_t'^{(i)} - c^{(i)})/\tau')}{\sum_{k=1}^K \exp((\tilde{c}_t'^{(k)} - c^{(k)})/\tau')} \tag{11}$$

where $\tau$ and $\tau'$ are the temperature for the student and teacher, respectively. $c$ is the center that is exponential moving average updated for more stable estimation. The multi-crop and local-to-global regularization are not applied in DePT. More details can be found in (Caron et al., 2021).

## B.3 EXTENDING TO MULTI-SOURCE DOMAIN TTA SETTING

The multi-source domain test-time adaptation setting aims at training a model from multiple source domains and adapt it to the unseen target domain. Many works aims at learning a domain-invariant feature across source domains. However, such methods are different to learn domain-invariant features if the source domain is diverse. They also can't capture domain-specific knowledge.

Our method is versatile to extend to multi-source domain test-time adaptation setting. See in Fig. 5, the basic idea is that all domains share a domain-agnostic backbone while each domain has its own prompts to learn the domain-specific knowledge. During source training, each domain prompts

along with a shared general prompts are randomly initialized. In each training iteration, we attractively sample images from each domain. The backbone, shared prompts and the corresponding domain prompts are optimized via supervised loss. During the TTA phase, prompts from all domain are averaged and concatenated with the shared prompts as the initialization. Then the optimization is the same as one-on-one TTA adaptation.

In our experiments, we use three domains in the DomainNet-126 as the source domains, and use the left one as the target.

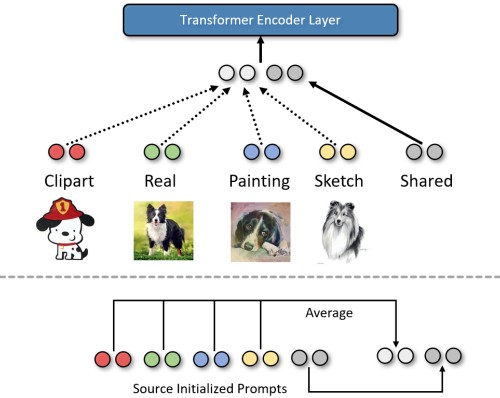

Figure 5: Our method is easy to extend to multi-source domain adaptation setting, where only the corresponding phase needs to be modified.

### B.4  IMPLEMENTATION DETAILS

We use PyTorch for all implementation. For source training, we initialized the ViT-B-16 with ImageNet pretrained weights from timm library (Wightman, 2019). The model name used in timm library is "vit_base_patch16_224", where the weight is pretrained in ImageNet21K with AugReg and then finetuned on ImageNet1K. We follow Chen et al. (2022) to randomly split the source data with 90% for training and 10% for validation. For source training, we train around 11k iterations. For target training, we train around 6k iterations for all datasets unless noted. For all experiments, we use SGD optimizer with momentum 0.9 and weight decay 1e-4, and cosine annealing on the learning rate. The training batch size is set as 128.

## C  MORE EXPERIMENT RESULTS

### C.1  ABLATION ON THE HYPER-PARAMETER CHOICE

Tab. 7 shows the impact of the hyper-parameter choice, including learning rate, the teacher temperature in the DINO loss, and the weight to balance training loss.

For visual prompt tuning, we find that a larger learning rate would benefit the performance. For example, learning rates of 1e-3 and 5e-3 lead to a decent performance. Continuing to amplify the learning rate may lead to instability in training.

The Teacher temperature is a hyper-parameter in the DINO loss that controls the sharpness of the teacher output distribution. A large temperature makes the teacher tend to output a uniform distribution, which makes the initial optimization difficult. The performance of small temperature is slightly worse. Using a warmup mechanism on the teacher's temperature is a good balance.

We finally investigate the effect of weights among the proposed losses in Eq. 6 . We set $\alpha$ to 1, and adjust the value of $\beta_1 = \beta_2 = \beta$. We set $\lambda = \beta/20$. We found that a small $\beta$ value results in better performance, e.g., below 0.1.

Despite the performance varies slightly by different hyper-parameter choices, DePT consistently demonstrates state-of-the-art performance compared to other methods, which shows the robustness of DePT against the choice of hyper-parameters.

Table 7: Ablation of the impact of hyper-parameter choices of DePT-G on VisDA-C.

| LR | 2e-4 | 5e-4 | 1e-3 | 5e-3 |
|---|---|---|---|---|
| Avg Acc | 89.6 | 90.0 | 90.4 | 90.6 |
| Teacher Temp | 0.04 | 0.05 | 0.07 | 0.04-0.07 |
| Avg Acc | 90.1 | 90.2 | 90.6 | 90.5 |
| $\beta$ | 0.01 | 0.05 | 0.1 | 0.5 |
| Avg Acc | 90.1 | 90.2 | 90.6 | 89.8 |

## C.2 COMPUTATION EFFICIENCY ANALYSIS

As we insert visual prompts into multiple level of the ViT, during test-time adaptation, the gradients need to be propagated all the way to the input for updating. Therefore, we investigate the computation cost in terms of training time and memory of different model modulation approaches, including tuning full model (AdaContrast), tuning feature extractor (SHOT), tuning layer normalization parameters (Tent) and three variants of DePT with the number of prompts 50 and 100, see in 8. The results is measured with the same loss (entropy minimization) on the VisDA-C dataset. We run for 500 iterations and report the average results. The input images has batch size 16 (input size: $16 \times 3 \times 224 \times 224$). All results are measured with one NVIDIA A10G GPU. DePT consumed less time and memory than AdaContrast and SHOT with 50 prompts, while consumed slightly more time and memory with 100 prompts.

Table 8: The comparison of training time, memory consumption and number of tunable parapmeters of different model modulation approaches. Measured with ViT-B backbone.

| | Full | Feature | LN | DePT-S50 | DePT-S100 | DePT-G50 | DePT-G100 | DePT-D50 | DePT-D100 |
|---|---|---|---|---|---|---|---|---|---|
| Time/s | 0.106 | 0.106 | 0.077 | 0.102 | 0.123 | 0.102 | 0.124 | 0.105 | 0.126 |
| Mem/M | 5133 | 5133 | 3955 | 4601 | 5149 | 4647 | 5207 | 4741 | 5323 |
| #Params/M | 85.81 | 85.80 | 0.038 | 0.048 | 0.086 | 0.16 | 0.32 | 0.47 | 0.94 |

## C.3 DETAILED EXPERIMENT RESULTS

We present the detailed experiments results on VisDA-C and DomainNet-126 dataset. The results of methods with ResNet backbone are presented.

Table 9: Detailed experiment results of Table 1. Classification accuracy (%) on VisDA-C train → val adaptation. Upper table: UDA methods. Middle table: TTA methods. Bottom table: online setting. All models with ViT-B backbone are reproduced by us, while the results with R101 backbone are cited from the original paper. Bold and underline denote the first and second high results.

| Method | Backbone | plane | bcycl | bus | car | horse | knife | mcycl | person | plant | sktbrd | train | truck | Avg. |
|---|---|---|---|---|---|---|---|---|---|---|---|---|---|---|
| DANN | R101 | 81.9 | 77.7 | 82.8 | 44.3 | 81.2 | 29.5 | 65.1 | 28.6 | 51.9 | 54.6 | 82.8 | 7.8 | 57.4 |
| CDAN | R101 | 85.2 | 66.9 | 83.0 | 50.8 | 84.2 | 74.9 | 88.1 | 74.5 | 83.4 | 76.0 | 81.9 | 38.0 | 73.9 |
| CAN | R101 | 97.0 | 87.2 | 82.5 | 74.3 | 97.8 | 96.2 | 90.8 | 80.7 | 96.6 | 96.3 | 87.5 | 59.9 | 87.2 |
| SWD | R101 | 90.8 | 82.5 | 81.7 | 70.5 | 91.7 | 69.5 | 86.3 | 77.5 | 87.4 | 63.6 | 85.6 | 29.2 | 76.4 |
| MCC | R101 | 88.7 | 80.3 | 80.5 | 71.5 | 90.1 | 93.2 | 85.0 | 71.6 | 89.4 | 73.8 | 85.0 | 36.9 | 78.8 |
| Source | R101 | 57.2 | 11.1 | 42.4 | 66.9 | 55.0 | 4.4 | 81.1 | 27.3 | 57.9 | 29.4 | 86.7 | 5.8 | 43.8 |
| Tent | R101 | 86.5 | 23.0 | 79.8 | 51.9 | 78.3 | 17.6 | 87.6 | 64.2 | 79.9 | 23.6 | 63.9 | 1.1 | 57.3 |
| SHOT | R101 | 95.3 | 87.5 | 78.7 | 55.6 | 94.1 | 94.2 | 81.4 | **80.0** | 91.8 | 90.7 | 86.5 | **59.8** | 83.0 |
| AdaCon | R101 | 97.0 | 84.7 | 84.0 | 77.3 | 96.7 | 93.8 | 91.9 | **84.8** | 94.3 | 93.1 | 94.1 | 49.7 | 87.2 |
| Source | ViT-B | 99.0 | 69.1 | 78.2 | 71.9 | 87.7 | 58.4 | 96.4 | 35.0 | 52.7 | 92.5 | 96.0 | 17.6 | 71.2 |
| Tent | ViT-B | 99.0 | 76.9 | 79.2 | 80.8 | 93.9 | 84.2 | 95.9 | 54.5 | 74.6 | 92.9 | 95.6 | 22.9 | 79.2 |
| SHOT | ViT-B | **99.5** | 91.8 | 88.7 | 65.1 | 98.6 | **98.0** | 96.0 | 66.1 | 95.1 | **98.9** | 96.8 | 52.4 | 87.3 |
| AdaCon | ViT-B | **99.5** | 94.2 | 91.2 | 83.7 | 98.9 | 97.7 | **96.8** | 71.5 | 96.0 | 98.7 | 97.9 | 45.0 | 89.2 |
| DePT-S (Ours) | ViT-B | 98.7 | 90.1 | 87.4 | 70.7 | 98.9 | 96.0 | **96.8** | 75.2 | 91.9 | 96.2 | 96.2 | 52.5 | 87.7 |
| DePT-G (Ours) | ViT-B | 99.2 | 93.0 | 93.4 | 85.6 | 99.4 | 98.6 | 96.7 | 76.2 | 98.2 | 97.9 | 96.6 | 52.0 | 90.6 |
| DePT-D (Ours) | ViT-B | 99.4 | 93.8 | **94.4** | **87.5** | **99.4** | **98.0** | 96.7 | 74.3 | **98.4** | 98.5 | 96.6 | 51.0 | **90.7** |
| Online Setting | | | | | | | | | | | | | | |
| AdaCon | ViT-B | 95.4 | 75.2 | 83.4 | **64.2** | 95.8 | 93.2 | 92.4 | 65.3 | 92.6 | 82.9 | **95.1** | 39.7 | 81.3 |
| DePT-G (Ours) | ViT-B | **98.3** | **87.6** | **88.9** | 62.1 | **98.4** | **95.2** | **95.9** | **67.1** | **95.2** | **97.3** | 94.9 | **50.1** | **85.9** |

Table 10: Detailed experiment results of Table 3. Classification accuracy (%) of seven domain shifts on DomainNet-126. All models with ViT-B backbone are reproduced by us, while the performance of other baselines with R50 backbone are cited from the original paper. Bold and underline denote the first and second highest results.

| Method | Backbone | R→C | R→P | P→C | C→S | S→P | R→S | P→R | Avg. |
|---|---|---|---|---|---|---|---|---|---|
| Source | R50 | 55.5 | 62.7 | 53.0 | 46.9 | 50.1 | 46.3 | 75.0 | 55.6 |
| TENT | R50 | 58.5 | 65.7 | 57.9 | 48.5 | 52.4 | 54.0 | 67.0 | 57.7 |
| SHOT | R50 | 67.7 | 68.4 | 66.9 | 60.1 | 66.1 | 59.9 | 80.8 | 67.1 |
| AdaContrast | R50 | 70.2 | 69.8 | 68.6 | 58.0 | 65.9 | 61.5 | 80.5 | 67.8 |
| Source | ViT-B | 68.7 | 75.9 | 69.8 | 67.7 | 74.1 | 60.4 | 86.4 | 71.8 |
| Tent | ViT-B | 69.1 | 77.7 | 70.1 | 67.0 | 75.3 | 61.7 | 88.0 | 72.7 |
| SHOT | ViT-B | 80.2 | 81.5 | 79.8 | 74.2 | **82.2** | 72.8 | 90.3 | 80.1 |
| AdaContrast | ViT-B | 81.8 | 81.8 | 82.0 | 75.8 | **82.2** | 73.7 | **90.4** | 81.1 |
| DePT-S (Ours) | ViT-B | 81.2 | 81.4 | 80.9 | 75.1 | 81.7 | 73.1 | 88.9 | 80.3 |
| DePT-G (Ours) | ViT-B | **83.3** | 81.6 | 82.6 | **77.8** | 81.3 | **75.1** | 89.9 | **81.7** |
| DePT-D (Ours) | ViT-B | 82.8 | **82.5** | **82.8** | 76.9 | 81.2 | 74.6 | 89.8 | 81.5 |
| Online Setting | | | | | | | | | |
| AdaContrast | ViT-B | 72.1 | 77.9 | 71.8 | 70.1 | 75.7 | 65.5 | **87.9** | 74.2 |
| DePT-G (Ours) | ViT-B | **74.8** | **78.3** | **74.2** | **72.5** | **77.2** | **68.5** | 87.2 | **76.1** |

