# OpenReview forum: "Visual Prompt Tuning For Test-time Domain Adaptation"
_ICLR.cc/2023/Conference — Submitted to ICLR 2023_

### Official Review · Reviewer_n2AM · 2022-10-23

**Confidence:** 4
**Correctness:** 3
**Technical Novelty And Significance:** 2
**Empirical Novelty And Significance:** 2
**Recommendation:** 5

**Clarity, Quality, Novelty And Reproducibility:**


Clarity and Quality: This paper is well-written and easy to follow. The description of the proposed method, the figures and the tables are clear.

Novelty: The novelty of the proposed memory-based PL and prompt tuning are somewhat limited. However, the diversity term is interesting.

Reproducibility: No source code is provided.



**Strength And Weaknesses:**


Strength

+ This paper is well-written and easy to follow. The motivation is clear and reasonable.

+ Clear figures are provided to help us better understand the details of the proposed method.

+ The proposed diversity term is novel and interesting.

+ Extensive experiments are provided to evaluate the effectiveness of the proposed method.

Weakness

+ The novelty of the proposed method is somewhat limited. The proposed method includes two parts, memory-based pseudo-labeling and visual prompt learning [A]. However, both parts are already studied in the domain adaptation community, especially memory-based pseudo-labeling.

+ Although the authors argue that the proposed method achieves significant improvement, the improvement is mainly obtained by pseudo-labeling (Table 5). The CLS regression can also improve the performance, but it is largely bought from DINO, i.e., distillation with teacher-student strategy. From Table 5, we can find that the improvements of the proposed memory-based PL and prompt tuning are somewhat limited.

+ To verify the effectiveness of prompt tuning, it is better to show the ablation study of each component in more datasets.

+ Although the proposed method requires few parameters to be tuned compared to other methods, I think the computation cost and training time are not reduced too much. This is because the prompts are injected into different layers, and the gradients should be calculated on all layers. In other words, tuning fewer parameters does not mean saving computation costs and training time. I thus think it is required to provide the computation cost (memory and training time) for different methods.

+ How to ensure that the injected prompts could contain domain-specific information? In addition, it would help us better understand the learned prompts by visualization as shown in Figure 3.

+ In the recent community, the test-time domain adaptation commonly refers to the online setting. While the experiments mostly conducted in this paper should be called source-free domain adaptation or test-time training adaptation. Please clarify them in the next version.

[A] Domain Adaptation via Prompt Learning.


**Summary Of The Paper:**

This paper considers the problem of test-time domain adaption. Specifically, the authors propose a data-efficient prompt tuning (DePT) approach, which only requires finetuning injected visual prompts and classifiers during training. In addition, a memory-based pseudo-labeling method is also introduced to improve the quality of pseudo-labels. Experiments on several datasets show the benefit of the proposed DePT.



**Summary Of The Review:**

This paper is well-written and easy to follow.  However, the novelty and improvement of the proposed method are limited. The computation cost is not evaluated to show the real benefit of the proposed method compared to other methods. Also, this paper lacks a convincing explanation for the prompt tuning. Considering these factors, I tend to give borderline reject.

---

> ### Author Response · Authors · 2022-11-17
> **First response to reviewer n2AM**
>
> Thank you for your kind feedback. We have carefully revised the paper following the suggestions. The following are our responses.
>
> 1. >"*The novelty of the proposed method is somewhat limited. The proposed method includes two parts, memory-based pseudo-labeling and visual prompt learning [A]. However, both parts are already studied in the domain adaptation community, especially memory-based pseudo-labeling.*"
>
> As far as we know, our method is the first to apply visual prompt tuning in the test-time adaptation task. Although some recent works attempt to use prompt learning for domain adaptation, they are usually highly dependent on the large pre-trained vision language model. Take paper [A] for example, they use pre-trained CLIP as the backbone, where a text encoder is required. In contrast, our DePT learns the visual prompts directly without relying on the text encoder. Moreover, paper [A] uses the UDA setting, while our DePT lies in the more challenging TTA setting. DePT proposes a novel and flexible way to modulate the model for adaptation.
>
> Pseudo-labeling is a well-studied approach, but it is essential in the test-time adaptation task. We do not overemphasize our contributions to pseudo-labeling. Actually, the pseudo-labeling part only takes less than half a page in the manuscript. Presenting the pseudo labeling method is needed for the completeness of the paper.
>
> From a technical point of view, 1. DePT proposes three variants to insert prompt to vision Transformer, which can adapt the representation of the fixed backbone to the target domain with excellent parameter efficiency and data efficiency. 2. We propose a hierarchical fine-grained self-supervised regularization for the prompt to avoid error accumulation in pseudo labeling. 3. A diversity term is proposed to increase the diversity of the prompt. These novelties are not proposed in the previous prompt-tuning works. The effectiveness of DePT is validated by SOTA performance on multiple datasets and diverse TTA settings.
>
> Moreover, the low data regime is one of our key novelties, but it seems to have been overlooked by the reviewers. We emphasize that it is an important and practical setting, yet hardly discussed in previous literatures. The previous source-free adaptation methods usually assume that the quantity of unlabeled target domain data is sufficient. However, in real-world tasks, even unlabeled data have high collection costs, including money and time. A method that can achieve high adaptation performance with limited target domain data will substantially reduce the cost.
>
> 2. >"*Although the authors argue that the proposed method achieves significant improvement, the improvement is mainly obtained by pseudo-labeling (Table 5). The CLS regression can also improve the performance, but it is largely bought from DINO, i.e., distillation with teacher-student strategy. From Table 5, we can find that the improvements of the proposed memory-based PL and prompt tuning are somewhat limited.*"
>
> We apologize that we did not clarify this in the previous version manuscript and confused you. The ablation reported in the previous version is under 100% data with DePT-G on VisDA-C. Since the 100% VisDA-C target domain has 55k images, this ablation cannot fully reflect the effects of each loss, so we added the ablation under 1% data and the results of tuning all parameters in the revised manuscript, see Tab. 6 (previous Tab. 5).
>
> With only pseudo labeling as the loss, the adaptation performance is improved a lot under 100% data (+12.4%). However, the improvement of 1% data is much less (+5.2%). We had the following observation during the TTA process. Under 1% data,  the classification accuracy of the model for pseudo-labels quickly increased to more than 99% and converged close to 100%. However, with 100% data, the classification accuracy of the model for pseudo-labels increases slowly and converges to about 92%. This phenomenon indicates that under 1% data, due to the limited amount of data, the model easily overfits to the wrong pseudo-label and gets more and more confident about its wrong prediction, thus having limited adaptation improvement.
> After adding self-supervised loss (CLS regularization, prompt regularization and prompt diversity loss), this error accumulation phenomenon has been greatly alleviated. These experiments show the effectiveness of the proposed self-supervised losses.
>
> We provide another comparison to show the effect of prompt tuning. In all the results mentioned above, we use DePT-G and only tune the prompt during TTA. The lower part of Table 6 provides the results of tuning all parameters (including prompts and backbone) with the same loss. Due to the huge amount of tunable parameters, the adaptation works well when there is enough unlabeled data (100%). However, under limited data (1%), overfitting to the noisy pseudo label happened again. Even adding the proposed self-supervised regularization cannot mitigate such overfitting.

---

> > ### Author Response · Authors · 2022-11-17
> > **Second response to reviewer n2AM**
> >
> > 3. >"*To verify the effectiveness of prompt tuning, it is better to show the ablation study of each component in more datasets.*"
> >
> > We provide one more ablation study under 1% data on VisDA-C. As ImageNet-C and DomianNet-126 have multiple domains, running ablation on them takes time. We are working on this. We’ll provide more ablations on these two datasets if the results come out before the rebuttal is due.
> >
> > 4. >"*Although the proposed method requires few parameters to be tuned compared to other methods, I think the computation cost and training time are not reduced too much. This is because the prompts are injected into different layers, and the gradients should be calculated on all layers. In other words, tuning fewer parameters does not mean saving computation costs and training time. I thus think it is required to provide the computation cost (memory and training time) for different methods.*"
> >
> > We agree that prompt tuning needs to compute the gradients all the way to the input for updating. We would also like to clarify that in the paper, we claim DePT is parameter-efficient and data-efficient in our manuscript rather than computation-efficient; it is worth noting that parameter-efficient (i.e., the model finetunes fewer parameters and therefore is less susceptible to overfitting) is different from computation-efficient (i.e., the model runs faster during training/inference).
> >
> > Following your suggestion, we added experiments about the computation cost in terms of training time and memory. We measured the training time for one iteration, memory consumption, and the number of parameters of different modulation methods with the same loss (entropy minimization) on the VisDA-C dataset. The compared methods are AdaContrast (Full), SHOT (Feature), TENT (LN), and three variants of DePT with the number of prompts 50 and 100. The numbers reported were measured with an average of 500 iterations with batch size 16 (input size: 16*3*224*224) on one NVIDIA A10G GPU. The results are presented in Appendix Table 8. DePT consumed less time and memory than AdaContrast and SHOT with 50 prompts, while it consumed more time and memory with 100 prompts.
> >
> > 5. >"*How to ensure that the injected prompts could contain domain-specific information? In addition, it would help us better understand the learned prompts by visualization as shown in Figure 3*"
> >
> > For one-to-one domain adaptation, the prompts are first trained with labeled source domain data with the backbone. The prompts are trained to minimize the loss in the source domain, thus learning source domain knowledge. During test-time adaptation on the target domain, only the prompt and the classification head is trained, thus the prompts learn target domain knowledge to adapt the model’s representation via self-attention layers.
> > For multi-source domain adaptation, we have a set of prompts for each source domain. In each source training iteration, we sample image batches from each domain. The backbone and the corresponding domain prompts are optimized via supervised loss. Thus the prompts can learn the domain-specific knowledge from the corresponding domain.
> >
> > 6. >"*In the recent community, the test-time domain adaptation commonly refers to the online setting. While the experiments mostly conducted in this paper should be called source-free domain adaptation or test-time training adaptation. Please clarify them in the next version.*"
> >
> > Test-time adaptation can be seen as a special setting of source-free adaptation, where the key is to adapt without access to the source domain data. Following TENT and AdaContrast, test-time adaptation has offline and online settings. The offline test-time adaptation allows the model to update multiple epochs with the target domain data before making an inference, which is equivalent to source-free adaptation. Online test-time adaptation setting has a more restrictive and challenging constraint than usual source-free adaptation. It assumes the target domain test data comes as a stream, where the model needs to update and inference at the same time batch-by-batch as long as there are testing data.
> >
> > As DePT demonstrates SOTA performance in both offline (even with limited data) and online settings, we prefer to name DePT as a test-time adaptation method, instead of source-free adaptation. We also made modifications in the introduction and related work sections to clarify the settings.

---

> ### Author Response · Authors · 2022-12-05
> **Followup response to reviwer n2AM**
>
> Dear reviewer n2AM:
>
> We would like to thank you again for your queries. In our responses, we have made every effort to answer your queries about the ablation study and the contribution of each component. We have also attempted to compare the computation cost with other methods. Moreover, the setting (TTA v.s. Source-free) is clarified,  missing citations are discussed, and other details are also provided. We eagerly await your further response to our replies, and we welcome any suggestions for improving our effort.
>
> Best.

---

### Official Review · Reviewer_H4uU · 2022-10-25

**Confidence:** 4
**Correctness:** 4
**Technical Novelty And Significance:** 2
**Empirical Novelty And Significance:** 2
**Recommendation:** 5

**Clarity, Quality, Novelty And Reproducibility:**

- the paepr is generally easy to understand.
- The paper does not propose any novel theory  but rather gathers ideas from other places and applies them to TTA which is okay.

in terms of reproducibility and robustness, its not clear how the hyper-parameters for equation 6 (the loss function) where chosen?. Do we assume we have access to a targe validation set (sebset of the target data with labels)

another question about clarity is the experimental setup for figure 1. I assume in figure 1 , the size of the test data is different for points in the horizantal axis and thus they arent exactly comparable.

Error bars (repeated experiments with different parameter initialization seeds) are missing from expeirments which makes it hard to evaluate reproducibility and robustness.

Whats the difference between strong transform and weak transform? Has there been any ablation study or discussion on why strong transform is used in some places and weak transform in other places?

**Strength And Weaknesses:**

strengths:
- The paper is easy to read and the flow of the paper is generally good.
- Figure 1 is highlights an important and intersting experiment with the number of available target examples in the semi-supervised learning setting.
- Results are very promising

weaknesses:
- I think there needs to be discussion and ablation study on the importance of visual prompts. They describe how they are trained but whats missing is a discussion on why there are needed.
- The paper misses some important citations. For example "hypothesis disparity regularized mutual information maximization" AAAi 2020 uses ensembles for regularization of TTA which is closely related to the proposed method.
- There needs to be more dicussion on what the self-supervised learning is set to achieve differently from that of the pseudo label training. Also what was the process of desing choice for selecting DINO over other SSL methods?
-Error bars (repeated experiments with different parameter initialization seeds) are missing from expeirments which makes it hard to evaluate reproducibility and robustness.


**Summary Of The Paper:**

The paper present a method for test time adaptation using learnable visual prompts integrated with visual transformers.  The paper uses various regularization methods to avoid error accomulation; knowledge distilation from ensemble or sourch checkpoints, self-supervised learning and learnable visual prompts.
The results are very promising.

**Summary Of The Review:**

The paper is interesting however, it misses some references (see above) and I cant seem to find an original point for novelty but the experimental results show significant improvement over the current state of the art. I hope the authors provide clarifications in the rebuttal phase.

---

> ### Author Response · Authors · 2022-11-17
> **First response to reviewer H4uU**
>
> Thank you for your kind feedback. We have carefully revised the paper following the suggestions. The following are our responses.
>
> 1. >"*I think there needs to be discussion and ablation study on the importance of visual prompts. They describe how they are trained but whats missing is a discussion on why there are needed.*"
>
> We discussed the motivation for visual prompts in the previous version, but it was left at the last of the appendix. In the revised version, we rearranged the order of the appendix and put the motivation at the start, see in Appendix A.
>
> How to modulate the model is vital for the TTA setting. Previous studies usually modulate the normalization layers or all parameters. However, current unsupervised learning objectives are not always accurate and usually contain noise and errors. If there are too many parameters to modulate, the current over-parameterized models are prone to overfit the noise of the unsupervised objective, especially when the amount of unlabeled target data is limited. On the other hand, if modulating too few parameters, although overfitting will be avoided, the adaptation capacity of the model will be limited. We verified the above hypothesis through two experiments, see Fig 4. Tuning prompt has the best trade-off between adaptation performance and overfitting risk.
>
> 2. >"*The paper misses some important citations. For example "hypothesis disparity regularized mutual information maximization" AAAi 2020 uses ensembles for regularization of TTA which is closely related to the proposed method.*"
>
> The mentioned work is cited and discussed in the related work section.
>
> 3. >"*There needs to be more dicussion on what the self-supervised learning is set to achieve differently from that of the pseudo label training.*"
>
> The pseudo label of unlabeled target domain data is predicted by the source-initialized model. The larger the domain gap, the lower the accuracy. Self-training with the wrong pseudo labels will make the model more and more confident about the wrong classification. We use a memory bank to refine the pseudo labels, where the top-k nearest neighbor in the memory bank is used to soft vote the class label. Voting with multiple data points usually results in a more robust classification, but it cannot completely eliminate false pseudo-labels. The accumulation of errors prevents the accuracy from improving. This phenomenon is more serious when the amount of target domain data is small, see Table 6, row 1.
>
> The goal of self-supervised learning is to let the representation of similar data be close in the representation space, while different data are apart. Besides, directly improving the representation quality of the model in the target domain, the self-supervised objective is also complementary to the pseudo-labeling. The self-supervised objective can encourage data from the same class to form a cluster, thus increasing the probability of correcting the wrong labels during soft voting and improving the pseudo-label quality. The effect of self-supervised learning is especially important given limited data, see Table 6, row 3, 4,  and 5.
>
> 4. >"*Also what was the process of desing choice for selecting DINO over other SSL methods?*"
>
> Among SSL methods, MoCo SimCLR needs negative samples for contrastive learning. In contrast, BYOL and DINO push the representation of positive pairs to be closer, which is more straightforward, where no negative samples are needed. DINO is designed for ViT, and its simple framework facilitates DePT to apply hierarchical fine-grained regularization on the aggregated prompts. Thus we adopted DINO as the objective. In fact, we have tried to use MoCo as the objective, but its performance is slightly lower than DINO (89.1 vs. 89.3 on VisDA-C), and it isn't easy to combine with prompts.
>
> 5. >"*Error bars (repeated experiments with different parameter initialization seeds) are missing from expeirments which makes it hard to evaluate reproducibility and robustness.*"
>
> We run three random seeds on 100\% VisDA-C with DePT-G, the results are 90.49, 90.61, 90.49. Therefore, the mean and std of these three runs are 90.53+-0.06. DePT shows stable performance under different random seeds.
>
> 6. >"*in terms of reproducibility and robustness, its not clear how the hyper-parameters for equation 6 (the loss function) where chosen?. Do we assume we have access to a targe validation set (sebset of the target data with labels)*"
>
> In fact, we did not spend a lot of time on the hyperparameter selection. In Table 7 of the Appendix, we can see that DePT is not sensitive to the selection of hyperparameters. A wide range of hyperparameters can achieve good results.

---

> > ### Author Response · Authors · 2022-11-17
> > **Second response to reviewer H4uU**
> >
> > 7. >"*another question about clarity is the experimental setup for figure 1. I assume in figure 1 , the size of the test data is different for points in the horizantal axis and thus they arent exactly comparable.*"
> >
> > The test data of different data points in Figure 1 are consistent, and they are all 100% test set. The difference between them is that the data used for TTA training are different. We use randomly sampled 1%, 10%, 50%, and 100% data respectively.
> >
> > 8. >"*Whats the difference between strong transform and weak transform? Has there been any ablation study or discussion on why strong transform is used in some places and weak transform in other places*"
> >
> > For strong augmentation, we use the augmentation in DINO, which consists of random resize, random crop, flip, color jitter, and gaussian blur. For weak augmentation, it only contains random resize, crop and flip. The weak augmented image is used to predict the pseudo labels. The model is trained to make the prediction of the strong augmented image to match the weak augmented one. This is because the model is more likely to get an accurate pseudo label on the weakly augmented image. Such design is commonly used in semi-supervised learning literatures, e.g.  UDA (Unsupervised data augmentation for consistency training), ReMixMatch and FixMatch.

---

> ### Author Response · Authors · 2022-12-05
> **Followup response to reviewer H4uU**
>
> Dear reviewer H4uU:
>
> We would like to thank you again for your queries. In our responses, we have made every effort to answer your queries about the motivation for using visual prompt tuning for test-time adaptation. We have also attempted to clarify the reason we choose DINO over other SSL methods. Moreover, the missing citations are discussed, the error bar is provided, and other details are also provided. We eagerly await your further response to our replies, and we welcome any suggestions for improving our effort.
>
> Best.

---

### Official Review · Reviewer_Za8v · 2022-10-26

**Confidence:** 4
**Correctness:** 3
**Technical Novelty And Significance:** 3
**Empirical Novelty And Significance:** 2
**Recommendation:** 5

**Clarity, Quality, Novelty And Reproducibility:**

- The paper is generally well written and easy to understand (except that some concepts need better explanation at their early appearance, e.g., the memory-based online pseudo labeling).

- Prompt tuning is a popular technique. However, using prompt tuning for test-time domain adaptation has some novelty.


**Strength And Weaknesses:**

**Strength**

- Using an efficient fine-tuning technique, i.e., prompt tuning for test-time adaptation is technically sound.

- The achieved accuracy is competitive.


**Weakness**

- Introduction for some important components lacks intuition and is not easy to understand, e.g., the memory-based online pseudo labeling in the abstract and the introduction. It not easy to understand its name and how it performs until the detailed method.

- While the training objectives are modified from recent self-supervised learning techniques, the authors have not provided necessary comparison or revisit. For example, the pseudo labeling part can be viewed as a modification from MoCo, but the authors introduce the momentum teacher as if it was invented for the first time. Moreover, the MSE losses in Eq. 4~5 (and their input features) have strong relation to BYOL.

- Based on the above concern, it is important to explain why the proposed method makes those modifications (instead of directly applying the original MOCO or BYOL).



**Summary Of The Paper:**

This paper proposes a test-time adaptation method based visual prompt tuning. After training the deep model along with a set of prompts to convergence, the proposed DePT fixes the backbone parameters and only fine tunes the prompts on the target domain. Based on this learning scheme, DePT integrates multiple self-training techniques (with adequate modification) into the training objective, including a pseudo labeling under the student-teacher framework, modified DINO, consistency on both the class token output (through cross-entropy loss) and the prompt token output (through mean squared error). Experimental results validate the effectiveness and investigate the major components through ablation.

**Summary Of The Review:**

I am generally satisfied with the contribution of this paper. I would like to raise my rating if the authors could address my concerns above, especially the relation between their training objectives and the original self-supervision methods.

---

> ### Author Response · Authors · 2022-11-17
> **First response to reviewer Za8V:**
>
> Thank you for your kind feedback. We have carefully revised the paper following constructive suggestions. The following are our responses.
>
> 1. >"*Introduction for some important components lacks intuition and is not easy to understand, e.g., the memory-based online pseudo labeling in the abstract and the introduction. It not easy to understand its name and how it performs until the detailed method*"
>
> DePT first predicts a pseudo label of the target sample. The initial pseudo label might contain errors due to the domain shift. To improve the quality of the pseudo labels,  we perform soft voting with the nearest neighbor data points in the memory bank. The memory bank is maintained by the EMA teacher and is updated online with the future target data batches.
> We have added more explanation of the memory-based online pseudo-labeling in the introduction section. Nevertheless,  due to the space limitation, details are described in the method section and Appendix B.1.
>
>
> 2. >“*While the training objectives are modified from recent self-supervised learning techniques, the authors have not provided necessary comparison or revisit. For example, the pseudo labeling part can be viewed as a modification from MoCo, but the authors introduce the momentum teacher as if it was invented for the first time. Moreover, the MSE losses in Eq. 4~5 (and their input features) have strong relation to BYOL*”
>
> Our learning objective mainly consists of two parts: pseudo labeling and self-supervised regularization. For pseudo labeling, it has been widely studied in semi/self-supervised learning. We did not overclaim that we invented the momentum teacher. Instead, we stated the student-teacher model we used is inspired by Mean Teacher and FixMatch at the start of section 3.2.1.
>
> The self-supervised regularization is introduced to mitigate error accumulation during self-training. Among MoCo, BYOL and DINO, MoCo needs to maintain a negative sample queue for contrastive learning. In contrast, BYOL and DINO push the representation of positive pairs to be closer, which is simpler and no negative samples are needed. DINO is designed for ViT, and its simple framework facilitates DePT to apply hierarchical fine-grained regularization on the aggregated prompts, thus we adopted DINO as the main objective.
>
> The way we add hierarchical fine-grained regularization on the aggregated prompts is similar to BYOL. This is because we find there is no need to apply DINO objective to the aggregated prompts. Adding centering and sharpening operations on the CLS token is enough to avoid collapse. Therefore, we directly minimize the MSE for simplicity. Moreover, we add a diversity term to increase the diversity of the prompts.
>
> 3. >"*Based on the above concern, it is important to explain why the proposed method makes those modifications (instead of directly applying the original MOCO or BYOL)*"
>
> Usually, contrastive learning is performed on the overall representation of the image, i.e. CLS token for ViT. The way we add prompts not only injects domain-related knowledge to the ViT, but also obtains a hierarchical and fine-grained representation. For example, the aggregated prompts at different positions of the ViT contain the corresponding levels of information, and each prompt focuses on different attributes of the image. Adding hierarchical fine-grained regularization can further improve the quality of representation on the target domain. For example, see in Table 6, adding regularization with prompt achieves higher performance than that with CLS token only.

---

> ### Author Response · Authors · 2022-12-05
> **Followup response to reviewer Za8v**
>
> Dear reviewer Za8v:
>
> We would like to thank you again for your queries. In our responses, we have made every effort to answer your queries about the intuition and details of memory-based pseudo labeling. We have also attempted to clarify the motivation and comparison of using different SSL objectives. We eagerly await your further response to our replies, and we welcome any suggestions for improving our effort.
>
> Best.

---

> > ### Comment · Reviewer_Za8v · 2022-12-07
> > **I have a similar concern with Reviewer H4uU on the novelty issue**
> >
> > After reading other reviewers' comments and the authors' responses, I have a similar concern with Reviewer H4uU regarding the novelty:
> >
> > 1) the fact that prompt tuning can achieve efficient fine-tuning effect is indeed already known by the community, while this paper merely employs its efficient fine-tuning characteristic as a tool.
> > 2) The real key points of this method, i.e., the pseudo labeling and the self-supervised regularization, are actually not necessarily tied with  the prompting technique (moreover, these key points are modified from existing unsupervised learning methods and are not considered as the novelty, either).
> >
> > I would like to listen to other reviewers' comments on this issue before making my final recommendation.

---

> > > ### Author Response · Authors · 2022-12-07
> > > **Novelty clarification for reviewer Za8V**
> > >
> > > We very much appreciate your time and the additional feedback. However, we have to disagree with some of your comments on novelty:
> > >
> > > 1. **Test-Time Adaptation.** We would like to emphasize that we are exploring the potential of prompt tuning in the **test-time adaptation** setting, which itself is an important and underexplored research problem with its own research community. Although there are some efficient fine-tuning works, they lie in a transfer learning setting, where labeled data from downstream tasks are used for supervised training. In contrast, we are in a more challenging test-time adaptation setting, which is unsupervised, as no target domain label is available. As far as we know, we are the first to explore prompt tuning in the test-time adaptation community. Moreover, test-time adaptation is an important, challenging, and practical setting in real-world deployment scenarios, as the distribution shift is inevitable.
> > >
> > > 2. **Hierarchical Visual Prompts.** Taking a step back, even in the transfer learning setting, visual prompt tuning is still a new technique that has not been fully studied. Take VPT (https://arxiv.org/abs/2203.12119) as an example, they proposed to tune the parameters of visual prompts with labeled downstream data, where the prompts are only considered as parameters. In contrast, our work shows that visual prompts are more than just parameters. For example, the aggregated prompts have the potential to serve as a hierarchy of fine-grained representation of the images. We also show that the diversity of visual prompts has a huge impact. This is why we disagree with the reviewer’s first point in the follow-up comments. We don’t merely employ prompt tuning as a tool; instead, we explore more potential of the prompts. Moreover, the proposed hierarchical fine-grained self-supervised regularization is highly tied to the visual prompts, which is why we disagree with the reviewer’s second point. Such topics have not been discussed in any other works.
> > >
> > > 3. **Low Data Regime.** We have one more contribution to the test-time adaptation community, i.e., substantial performance improvement in the low data regime. We emphasize the low data regime is an important setting yet has been ignored by previous works (also by the reviewers). As we explained in the 4th point in our response to reviewer zNVW, collecting a large unlabeled target domain data is very costly in some real-world applications, and therefore performance of test-time adaptation methods in the low data regime is an important evaluation metric that is worthy of attention. Our DePT shows superior performance in the low data regime compared to previous SOTA methods.

---

### Official Review · Reviewer_zNVW · 2022-10-27

**Confidence:** 4
**Correctness:** 3
**Technical Novelty And Significance:** 2
**Empirical Novelty And Significance:** 3
**Recommendation:** 6

**Clarity, Quality, Novelty And Reproducibility:**

*Clarity*:

- This work can be understood, but it could be more accessible. The exposition would be improved by making introductory statements more definite and citing established vocabulary when it is first used. There are grammatical errors that take time to read through, so proofreading would also make this work more readable.
- The introduction does not fully explain the setting. It should define the test-time adaptation setting and contrast it with source-free adaptation, as the two are sometimes confused, and the established unsupervised domain adaptation setting, which is often synonymous with "domain adaptation" in the literature.
- The summarization of the memory bank in Figure 2(B) does not adequately explain the aggregated prompt or how the contents of the memory bank are initialized and updated. The caption should explain the multiple losses, if only briefly, and indicate how the memory bank alters the pseudo-labels.
- Examples are relegated to the appendix, such as appendix B.1, which would better serve the reader as part of the main paper. B.1 in particular has key motivating results for visual prompts as parameters.
- Overall the writing, figures, and tables are adequate for communicating the method and results, with only a few exceptions.

*Quality*:

- The results improve on standard benchmarks, by as much as +4 points when comparing to the prir state-of-the-art and controlling for backbone, which is a larger margin than is sometimes reported in papers on test-time adaptation (where improvements may be +1 or +2 points).
- The method is sensible, as visual prompting has been shown to work elsewhere such as for transfer learning, and the experiments justify this choice of parameterization for the purpose of test-time adaptation.
- The related work could better credit other visual prompting work that has been done (see weaknesses), but at least the application contributed here for test-time adaptation is executed well and delivers an improvement.

*Novelty*:

- Test-time adaptation to shift is a novel application of visual prompts in the form of additional learnable tokens. The experiments on choices of parameterization for test-time adaptation are empirically novel and informative.
- The pseudo-labeling update (Section 3.2.1) is not novel, as moving statistics in the form of exponential moving averages and student-teacher updates are common for self-supervision (MoCo, ODIN, ...) and adaptation (SHOT, DINE, ...).
- The self-supervision update (Section 3.2.2) is essentially DINO, as cited, but it has a few of its own implementation details including a diversity regularizer across the leraned prompts.
- Adapting to smaller amounts of data (Figure 3) is not novel, as claimed in the abstract, because online test-time adaptation methods already only adapt to the data given for testing.
  The significance of its insensitivity to the amount of data for adaptation is also more theoretical than practical, as knowing when to adapt or not is itself a problem, which is why prior methods like TTT or Tent keep adapting.

*Reproducibility*:

The explanation and appendices have a fair amount of detail, but given the number of terms in the loss and variety of update schemes (teacher-student, online memory, EMA, etc.) this work may not be reproducible from the paper alone.
There is no statement about releasing the code.



**Strength And Weaknesses:**

*Strengths*:

- This work brings test-time adaptation up-to-date w.r.t. current model architectures and parameters: ViTs are current, strong models and prompting is a timely topic.
  The choice of parameterization is novel for this purpose, low-dimensional for efficiency, and specifically-chosen for ViTs.
- Online adaptation accuracy improves on the prior state-of-the-art AdaContrast by 4 points on VISDA-C and 2 points on DomainNet (which are both standard benchmarks).
  Offline adaptation accuracy is no worse than the state-of-the-art and perhaps slightly better.
- The choice of datasets is sound. VISDA-C is a standard benchmark, and DomainNet is a more recent benchmark that has also seen adoption. The evaluation protocol for these datasets follows prior work.
- There is a self-supervised loss that does not depend on the pseudo-labels, for bottom-up adaptation to the target inputs, which may complement top-down adaptation to the model predictions.
  This loss is closely-related to DINO, which is credited accordingly, but it is somewhat customized to the use of visual prompts in this work.

*Weaknesses*:

- While DePT efficient in its parameter dimension, that does not mean it is efficient in its update computation. For DePT, the prompts are distributed throughout the model, and so optimization has to compute many gradients across layers even if the parameter gradients are themselves much smaller.
  The amount of computation required during testing is not described, nor is it compared against baselines.
- There is only one comparable baseline with the same model architecture (AdaContrast). For thoroughness, it would be useful to adapt a prior method that reported results with ResNets to ViTs for comparison. Tent for instance simply minimizes the entropy of predictions for a given choice of parameters, and could be ported to ViTs by updating the layer norm parameters (for one example).
- There are no results for other common choices of robustness or domain generalization benchmark, such as ImageNet-C (or its other variations ImageNet-V2, ImageNet-R, etc.), or PACS/VLCS/etc. These are included in many prior works such as TENT, SHOT, and TTT.
- There is missing related work on learning and tuning visual prompts, although neither addresses the application of visual prompts to test-time adaptation against shifts, as done in this work.
  - [Interactive Image Segmentation via Backpropagating Refinement Scheme](https://vcg.seas.harvard.edu/publications/interactive-image-segmentation-via-backpropagating-refinement-scheme/paper). Jang & Kim, CVPR'19. This paper prefigures visual prompts by perturbing the input according to a model's own predictions as a kind of self-training to improve accuracy. Citing it would help better trace the history of ideas.
  - [Exploring Visual Prompts for Adapting Large-Scale Models](https://arxiv.org/abs/2203.17274). Bahng et al. arXiv'22. This is highly related to the cited VPT by Jia et al. 2022 and should be considered for reference alongside it.
- The proposed method is not fully test-time because the prompts must be jointly trained with the source model parameters. Its setting should be better identified in the text, and the experiments should compare to the latest source-free methods accordingly, like SHOT++ (Source Data-absent Unsupervised Domain Adaptation through Hypothesis Transfer and Labeling Transfer. Liang et al. PAMI'21.)



**Summary Of The Paper:**

Test-time adaptation updates a model to reduce generalization on shifted data.
Such adaptation methods need to choose a loss for adaptation and parameters to update, and this work's main contribution is to introduce visual prompts as a parameterization.
The proposed Data-efficient Prompt Tuning (DePT) method includes additional learnable tokens in a ViT during training, then updates these tokens and the classification token during testing, by entropy minimization through pseudo-labeling and other losses.
The adaptation parameters are limited to the classification head and the proposed layer-wise prompts, which strikes a balance between expressivity and efficiency, where the number of prompts can be varied from just one layer to all layers.
This parameterization is justified by the accuracy achieved (Table 1) and analysis experiments that try alternatives like updating layer normalization parameters (Appendix B.1).
The other losses include a self-supervised regularization scheme derived from DINO, in which projections of the prompts (local representations) and the class token (the global representation) are updated to maximize their similarity.
Experiments evaluate DePT against the prior state-of-the-art AdaContrast, reproduced with the same ViT backbone as the proposed method, and other test-time methods with ResNet backbones (which are therefore not comparable), along with unsupervised domain adaptation methods that require both source and target data.
DePT achieves higher accuracies by offline adaptation on the whole test set and online adaptation on the streaming test set, with a larger margin of improvement for online adaptation.
These results are shown for domain adaptation and generalization datasets, VisDA-C and DomainNet-126, but not corruption or other robustness benchmarks like ImageNet-C or ImageNet-R.
Ablation studies show that each component of the method helps.
Although pseudo-labeling makes the largest difference, and that is already a popular part of test-time adaptation, it is pseudo-labeling as a loss to update the proposed parameterization by prompts that is helping so much.



**Summary Of The Review:**

In short, the method does work as shown by its evaluation on the standard datasets of VisDA-C and DomainNet.
However, the evaluation could be broader in its datasets and more thorough in its comparable baselines by reproducing more than one method with the ViT backbone.
This is an expectation for work in this area, as many past test-time adaptation papers have evaluated on several datasets of shifts as a way to shown some measure of generality.
All in all, there is value in demonstrating that test-time adaptation applies to ViT models too, and in exploring the right choice of parameterization for this purpose.
While this work could be improved, pushing test-time adaptation to use the most accurate architectures at present may be sufficiently informative to the community.

*For Rebuttal*

1. Please measure the time efficiency of DePT updates, and relate the computation required to update the prompts to the computation required to update normalization layers or other choices of parameters, as used by the compared methods like TENT and SHOT.
2. Please discuss DePT as a source-free (SHOT) vs. test-time (TENT) vs. intermediate (TTT) method. The choice of setting should guide the choice of comparisons.
3. Please report results for ImageNet-C, if possible, because it is a common benchmark for robustness and adaptation to corruption. (It is alright if this is not possible due to computational limitations, but doing it would be a plus.)
4. Please motivate the low data regime reported in the results. Is there a practical deployment that this setting addresses?

*Other Feedback*

- Title
  - Is the method a test-time adaptation method? Test-time methods update during testing, as DePT does, but in some usages "test-time" also indicates that the method can update online and that it does not require changes to training. DePT seems to be more like SHOT or TTT in that it alters training.
- Abstract
  - Name the datasets, benchmarks, and settings evaluated. A precise summary of experiments is more eye-catching to the potential reader.
- Related Work
  - The formatting with subsections takes up a lot of space. Consider revising it to remove the subsection headings and use more bolded heading instead. Then the paper would have more room for further explanation or experiments.
- Analysis
  - Table 3 is not an impressive comparison, because DePT has been augmented to include domain-specific and domain-shared parameters while SHOT has not been. Other methods like SHOT could also have domain-specific layers, as done by residual adapters (Rebuffi et al.) for example.
  - Table 4 should include the accuracy without DePT in the caption to help ground the relative improvement of each number of prompts.
- Proofreading
  - "Second, given only unlabeled target domain data, use what kind of learning objective for optimization." Should this be a question? Please revise.

---

> ### Author Response · Authors · 2022-11-17
> **First response to reviewer zNVW:**
>
> Thank you very much for your kind feedback. We have carefully revised the paper following constructive suggestions. The following are our responses.
>
> 1. >"*Please measure the time efficiency of DePT updates, and relate the computation required to update the prompts to the computation required to update normalization layers or other choices of parameters, as used by the compared methods like TENT and SHOT.*"
>
> We agree that prompt tuning needs to compute the gradients all the way to the input for updating. We would also like to clarify that in the paper, we claim DePT is *parameter-efficient* and *data-efficient* in our manuscript rather than *computation-efficient*; it is worth noting that *parameter-efficient* (i.e., the model finetunes fewer parameters and therefore is less susceptible to overfitting) is different from *computation-efficient* (i.e., the model runs faster during training/inference).
>
> Following your suggestion, we added experiments about the computation cost in terms of training time and memory. We measured the training time for one iteration, memory consumption, and the number of parameters of different modulation methods with the same loss (entropy minimization)  on the VisDA-C dataset. The compared methods are AdaContrast (Full), SHOT (Feature), TENT (LN), and three variants of DePT with the number of prompts 50 and 100. The numbers reported were measured with an average of 500 iterations with batch size 16 (input size: 16\*3\*224\*224) on one NVIDIA A10G GPU. The results are presented in Appendix Table 8. DePT consumed less time and memory than AdaContrast and SHOT with 50 prompts, while it consumed more time and memory with 100 prompts.
>
> 2. >"*Discussion about the setting of source-free (SHOT) vs. test-time (TENT) vs. intermediate (TTT) method.” and “Is DePT a test-time adaptation method?*"
>
> We also add the main comparable baseline AdaContrast into the discussion. We first discuss the test-time and source-free adaptation, two similar but slightly different settings. Then we talk about the details of the related work.
>
> Test-time adaptation can be seen as a special setting of source-free adaptation, where the key is to adapt without access to the source domain data. Following TENT and AdaContrast, test-time adaptation has offline and online settings. The offline test-time adaptation allows the model to update multiple epochs with the target domain data before making an inference, which is equivalent to source-free adaptation. Online test-time adaptation setting has a more restrictive and challenging constraint that usual source-free adaptation does not have. Online test-time adaptation assumes the target domain test data comes as a stream, where the model needs to update and inference at the same time batch-by-batch as long as there are testing data. Each image can be seen by the model only once.
>
> TTT, TENT, and AdaContrast all have offline and online settings, so they can be considered test-time adaptation methods. The SHOT paper does not mention the online setting, so it can be considered a source-free adaptation method. As indicated by TENT, an adaptation that does not alternate training is named “fully” test-time adaptation. DePT needs to train the prompts in the source domain, so it is not a “fully” test-time adaptation method. As DePT demonstrates superior performance in both offline (even with limited data) and online settings, we prefer to name DePT as a test-time adaptation method, instead of source-free adaptation. We also modified the introduction and related work sections to clarify the settings.
>
> 3. >"*Please report results for ImageNet-C, if possible, because it is a common benchmark for robustness and adaptation to corruption.*"
>
> The result of ImageNet-C is added to the revised manuscript, see in Table 2 and Section 4.3. DePT-G achieves the least error for 14 out of 15 types of corruption in the offline setting. In the online adaptation, DePT-G outperforms the AdaContrast and CFA with 3.7% and 1.2%, respectively.

---

> > ### Author Response · Authors · 2022-11-17
> > **Second response to reviewer zNVW**
> >
> > 4. >"*Please motivate the low data regime reported in the results. Is there a practical deployment that this setting addresses?*" and "*Adapting to smaller amounts of data (Figure 3) is not novel, as claimed in the abstract, because online test-time adaptation methods already only adapt to the data given for testing. The significance of its insensitivity to the amount of data for adaptation is also more theoretical than practical, as knowing when to adapt or not is itself a problem, which is why prior methods like TTT or Tent keep adapting.*"
> >
> > The low data regime is one of our key novelties, but it has been overlooked by the reviewers. We emphasize that it is an critical and practical setting, yet hardly discussed in previous literatures. The previous source-free adaptation methods usually assume that the quantity of unlabeled target domain data is sufficient. However, even unlabeled data have high collection costs in some real-world tasks, including money and time. A method that can achieve high adaptation performance with limited target domain data will substantially reduce the cost. Here are some examples.
> > - The cost of acquiring satellite data is related to the area. Acquiring satellite images of a large area is expensive. (
> > - For medical images, the data quantity is limited due to the difficulty of data acquisition, privacy issues, etc. Also, the collection time is extremely long due to lengthy processes like desensitization, ethical review, etc.
> >
> > On the other hand, the online test-time adaptation setting looks promising, where the model keeps updating during testing. However, several factors limit it from lab research to practical application:
> > - The performance of Online TTA is also related to the amount of data. At the start of online TTA, its performance is low as it only sees a few target data. Online TTA  requires a certain period to update (more target data) to reach an acceptable performance. However, in the real-world product, users are very concerned about the product's initial performance when they first use it. Poor initial performance results in higher churn rates. Offline adaptation with a small amount of target data can help to alleviate this problem.
> > - Current online TTA methods are not theoretically guaranteed to avoid performance degradation or collapse. Unpredictable performance degradation or collapse in a real production environment will lead to disastrous consequences.
> > - Online training and backpropagation will reduce the inference speed. Moreover, model acceleration methods are usually not applicable to training. An additional cost is also required to support training in the deployment environment.
> >
> > All in all, the low data test-time adaptation is an important setting but has yet to be fully studied. Our DePT has good parameter and data efficiency, showing excellent performance in both offline (even with limited data) and online settings.
> >
> > 5. >"*There is only one comparable baseline with the same model architecture (AdaContrast). For thoroughness, it would be useful to adapt a prior method that reported results with ResNets to ViTs for comparison.*"
> >
> > We reproduced the Tent and SHOT with ViT-B backbone on the VisDA-C and DomainNet-126. The results has been added to the manuscript.
> >
> > 6. >"*There is missing related work on learning and tuning visual prompts, although neither addresses the application of visual prompts to test-time adaptation against shifts, as done in this work.*"
> >
> > The works are correctly cited and discussed in the related work section.
> >
> > 7. >"*The proposed method is not fully test-time because the prompts must be jointly trained with the source model parameters. Its setting should be better identified in the text, and the experiments should compare to the latest source-free methods accordingly, like SHOT++*"
> >
> > DePT is applicable for both offline (source-free) and online test-time adaptation. DePT needs to train the prompt with the source data. We clarify this in the introduction section. We compared the latest source-free methods with offline AdaContrast, which was published at CVPR2022. SHOT++ adds an additional step upon SHOT to transfer the label of high-confidence samples to less confident ones on the whole target domain via the semi-supervised learning algorithm MixMatch. Such an additional step is also applicable to other methods like DePT or AdaContrast. We prefer not to compare with SHOT++ for fairness.
> >
> > 8. >"*Other feedback*"
> >
> > We addressed all other feedback in the updated manuscript. Thanks for the suggestions! (a) We further clarify our setting (test-time adaptation vs. source-free adaptation) in the introduction and related work sections. (b) We add the benchmark and settings used in the abstract. (c)  We reformatted the related work section to save more room. The results of the ImageNet-C dataset are added to the experiments section. (d) We have done detailed proofreading.

---

> > > ### Comment · Reviewer_zNVW · 2022-11-29
> > > **Thank you for the thorough response.**
> > >
> > > The response and revision have addressed the main questions and weaknesses from the review including the points specifically highlighted for the rebuttal. The response and revision provide
> > >
> > > 1. more comparable baselines (SHOT + TENT with ViT-B/16),
> > > 2. more results on the corruptions benchmark (ImageNet-C) that is common for test-time adaptation,
> > > 3. more ablations to justify the prompt parameters and auxiliary losses proposed by DePT, and
> > > 4. more clarity in the exposition of the setting, method, and related work.
> > >
> > > As such I maintain my recommendation siding with acceptance. The purposes of this paper, in my estimation of its contributions to the community, is to show that prompts/tokens can serve as parameters for test-time adaptation, and to highlight test-time adaptation in the low data regime of early or limited access to shifted data. The revision has addressed the main flaws that counterbalance these contributions, namely the lack of comparable baselines and a common benchmark used by prior work.
> > >
> > > I encourage the authors to include the additional results and appendices in any published version, because they are informative, including the profiling of the computation (even though computation is not a specific claim of this paper).

---

> > > > ### Author Response · Authors · 2022-12-05
> > > > **Thank you for recognizing our contributions**
> > > >
> > > > Dear reviewer zNVW,
> > > >
> > > > We would like to thank you again for your detailed, constructive, and in-depth feedback and suggestions. They are really helpful in improving our manuscript. We also appreciate you for recognizing the contributions of our work to the community.
> > > >
> > > > Best.

---

### Author Response · Authors · 2022-11-17
**Rebuttal Summary**

Dear reviewers and area chair,

We thank you for your efforts and insightful suggestions. We followed the constructive feedback and carefully revised the manuscript. The main modifications include the following:
- We reproduced TENT and SHOT with ViT-B-16 backbone for comparison on the VisDA-C and DomainNet-126 datasets.
- We added experiments on the ImageNet-C dataset to evaluate the effectiveness of DePT against common image corruptions.
- We carefully revised the manuscript and appendix to add more clarification and discussion about the motivation, novelty, settings, related works, method details, and computation efficiency.
- We added more ablation studies to verify the contribution of each component.

Moreover, regarding the novelty of our article, we make an overall clarification here.

From the perspective of the test-time/source-free adaptation task, we have made two major contributions.
- We are the first to introduce visual prompt tuning to TTA setting. DePT proposes a novel and flexible way to modulate the model for adaptation. It can balance the adaptation capacity and the risk of performance degradation or collapse caused by overfitting. Appendix A discusses the advance of prompt tuning against commonly used full tuning and normalization layer modulation.
- We discussed the low data regime in the TTA setting, a critical and practical issue yet rarely discussed in the test-time/source-free adaptation community.  Even unlabeled data have high collection costs in some real-world tasks, including money and time. Low target domain data requirement enables fast and low-cost adaptation. DePT performs significantly better than previous literature on limited data.

From a technical point of view:
- DePT proposes three variants to insert prompt to vision Transformer, which can adapt the representation of the fixed backbone to the target domain with excellent parameter efficiency and data efficiency.
- We propose a hierarchical fine-grained self-supervised regularization for the prompt to improve target representation learning and avoid error accumulation in pseudo labeling.
- A diversity term is proposed to increase the diversity of the prompt. These novelties are not discussed in the previous prompt-tuning works.
- The effectiveness of DePT is validated by SOTA performance on three major adaptation datasets: VisDA-C, ImageNet-C, DomainNet-126, and diverse TTA settings including offline (even with limited data), online and multi-source domain adaptation.

The code will be released after acceptance. The point-to-point responses are posted to each reviewer. We hope these responses have addressed your concerns. Please let us know if you have further questions.

---

### Decision · Program_Chairs · 2023-01-20

**Decision:**

Reject

**Justification For Why Not Higher Score:**

This paper does not bring any new insight into the community. It simply deploys a method in a task. The weakness clearly outweighs the strength.

**Justification For Why Not Lower Score:**

N/A

**Metareview: Summary, Strengths And Weaknesses:**

This paper presents a test-time unsupervised domain adaptation (training unlabeled target data without access to the source data) method called DePT, based on visual prompt-tuning based ViT models. Experimental results show that the proposed DePT outperforms other ViT based methods.

Strength:
1. Presentation is clear
2. Applying visual prompt-tuning ViT in test-time UDA is novel

Weakness:
1. Though given Strength 2, the authors didn't provide any insights into why using such ViT models is especially useful for test-time UDA. For example, visual prompt is essentially a set of visual priors learned from ViT, why can such priors encode informative representations of source data?

2. The experimental comparisons are not carefully designed. For example, comparison with the traditional non-test time  UDA is non-informative; Ablations should be more emphasized on the visual prompt based ViT.

**Summary Of Ac-Reviewer Meeting:**

This a borderline paper with only one reviewer weakly supporting the paper. After rebuttal, the three negative reviewers are not convinced. To make a fair judgment, AC read the paper carefully and agree with the weakness raised by all the reviewers. AC thinks that the key concerns were not yet addressed by the reviewers, especially for the justification of why the visual-prompt ViT can be used in the test-time UDA task.